# Establishment and maintenance of motor neuron identity via temporal modularity in terminal selector function

Yinan Li[1,2], Anthony Osuma[1,2], Edgar Correa[1,3], Munachiso A Okebalama[1], Pauline Dao[1], Olivia Gaylord[4], Jihad Aburas[1], Priota Islam[5,6], André EX Brown[5,6], Paschalis Kratsios[1,2,3,4,7]*

[1]Department of Neurobiology, University of Chicago, Chicago, United States; [2]Committee on Neurobiology, University of Chicago, Chicago, United States; [3]Cell and Molecular Biology Program, University of Chicago, Chicago, United States; [4]Committee on Development, Regeneration and Stem Cell Biology, University of Chicago, Chicago, United States; [5]MRC London Institute of Medical Sciences, London, United Kingdom; [6]Institute of Clinical Sciences, Imperial College London, London, United Kingdom; [7]The Grossman Institute for Neuroscience, Quantitative Biology, and Human Behavior, University of Chicago, Chicago, United States

**Abstract** Terminal selectors are transcription factors (TFs) that establish during development and maintain throughout life post-mitotic neuronal identity. We previously showed that UNC-3/Ebf, the terminal selector of *C. elegans* cholinergic motor neurons (MNs), acts indirectly to prevent alternative neuronal identities (Feng et al., 2020). Here, we globally identify the direct targets of UNC-3. Unexpectedly, we find that the suite of UNC-3 targets in MNs is modified across different life stages, revealing 'temporal modularity' in terminal selector function. In all larval and adult stages examined, UNC-3 is required for continuous expression of various protein classes (e.g. receptors, transporters) critical for MN function. However, only in late larvae and adults, UNC-3 is required to maintain expression of MN-specific TFs. Minimal disruption of UNC-3's temporal modularity via genome engineering affects locomotion. Another *C. elegans* terminal selector (UNC-30/Pitx) also exhibits temporal modularity, supporting the potential generality of this mechanism for the control of neuronal identity.

*For correspondence:
pkratsios@uchicago.edu

Competing interests: The authors declare that no competing interests exist.

## Introduction

Nervous system development is a multi-step process that culminates in the generation of distinct neuron types necessary for animal behavior. Seminal studies in many model systems have begun to elucidate the molecular mechanisms that control the early steps of neuronal development, such as specification of neural progenitors and generation of post-mitotic neurons (*Catela and Kratsios, 2019*; *Doe, 2017*; *Greig et al., 2013*; *Jessell, 2000*; *Lodato and Arlotta, 2015*; *Perry et al., 2017*). However, our understanding of the final steps of neuronal development is very rudimentary. Once neurons become post-mitotic, how do they acquire their unique functional features, such as neurotransmitter synthesis, electrical activity and signaling properties? And, perhaps more importantly, how do neurons maintain such features throughout post-embryonic life?

Terminal selectors represent one class of transcription factors (TFs) with continuous expression – from development through adulthood – in specific neuron types (*Hobert, 2008*; *Hobert, 2011*; *Hobert, 2016b*; *García-Bellido, 1975*). A defining feature of terminal selectors is the ability to activate the expression of effector genes, whose protein products determine the terminally

differentiated state, and thereby function, of a given neuron type. Such effector genes, herein referred to as 'terminal identity genes', are expressed continuously in specific neuron types and endow them with distinct functional and phenotypic properties. Examples include neurotransmitter (NT) biosynthesis components, NT receptors, ion channels, neuropeptides, and adhesion molecules (*Hobert, 2008*; *Hobert, 2011*; *Hobert, 2016b*). Numerous studies support the idea that terminal selectors establish during development and maintain throughout life neuronal identity (and function) by activating expression of terminal identity genes (*Deneris and Hobert, 2014*; *Hobert, 2008*; *Hobert, 2016b*; *Hobert and Kratsios, 2019*). Multiple cases of terminal selectors have been described thus far in worms, flies, chordates and mice (*Deneris and Hobert, 2014*; *Hobert and Kratsios, 2019*; *Konstantinides et al., 2018*), suggesting high conservation of this type of regulators. Importantly, human mutations in terminal selectors and their effector target genes have been linked to either developmental or degenerative conditions of the nervous system (*Deneris and Hobert, 2014*).

However, the molecular mechanisms through which terminal selectors establish and maintain neuronal identity are poorly understood, in part due to two major challenges. First, the majority of studies follow a candidate approach focused on a select subset of terminal identity genes (*Flames and Hobert, 2009*; *Hobert, 2016a*; *Lopes et al., 2012*). Hence, the extent of terminal identity features and breadth of biological processes controlled by terminal selectors remain largely unknown. Addressing this knowledge gap requires unbiased methods for the identification of terminal selector target genes, but such approaches have only been applied to a limited number of terminal selectors to date (*Corbo et al., 2010*; *Housset et al., 2013*; *Kadkhodaei et al., 2013*; *Wyler et al., 2016*; *Yu et al., 2017*). Second, the continuous expression of terminal selectors represents an additional challenge because it is not known whether their function - in a particular neuron type - remains the same, or changes at different life stages. This is partly due to the difficulty of tracking individual neurons in the complex vertebrate nervous system throughout embryonic and post-natal life. Hence, longitudinal studies in simple model organisms are needed to determine whether terminal selectors control an identical suite of target genes across different stages (e.g., embryo, adult), or whether the suite of targets can change over time. Addressing these two challenges may extend our knowledge of how terminal selectors control neuronal identity, as well as advance our understanding of how cellular identity is established and maintained.

This study focuses on UNC-3, the sole *C. elegans* ortholog of the Collier/Olf/Ebf (COE) family of TFs (*Dubois and Vincent, 2001*). UNC-3 acts as a terminal selector in cholinergic motor neurons (MNs) of the *C. elegans* ventral nerve cord (*Kratsios et al., 2011*). Importantly, mutations in EBF3, a human ortholog of UNC-3, cause a neurodevelopmental syndrome characterized by motor developmental delay (*Blackburn et al., 2017*; *Chao et al., 2017*; *Harms et al., 2017*; *Sleven et al., 2017*). A previous study proposed that UNC-3 controls cholinergic MN identity in *C. elegans* by activating directly the expression of various terminal identity genes (e.g., acetylcholine biosynthesis components, ion channels, NT receptors, neuropeptides), which were identified via a candidate approach (*Kratsios et al., 2011*). More recently, it was demonstrated that UNC-3 can also act indirectly to prevent the adoption of alternative neuronal identities (*Feng et al., 2020*). Lastly, animals lacking *unc-3* gene activity display severe locomotion defects (*Brenner, 1974*; *Feng et al., 2020*), suggesting UNC-3 may broadly control gene expression in cholinergic MNs. However, an unbiased identification of UNC-3 targets, as well as a longitudinal analysis of *unc-3* mutants are currently lacking.

Here, we employ chromatin immunoprecipitation followed by DNA sequencing (ChIP-Seq) and report the identification of ~3500 protein-coding genes as putative direct targets of UNC-3. Protein class ontology analysis suggests that UNC-3, besides terminal identity genes, also controls additional biological processes, such as neuronal metabolism and downstream gene regulatory networks comprised of numerous TFs and nucleic acid-binding proteins. These findings help obtain a comprehensive understanding of terminal selector function.

Through a longitudinal analysis of *unc-3* mutants at embryonic, larval and adult stages, we identified two groups of target genes with distinct temporal requirements for UNC-3 in cholinergic MNs. One group encodes multiple classes of proteins (e.g., receptors, secreted molecules, TFs) that require UNC-3 for both embryonic initiation and post-embryonic maintenance of their expression.

Contrasting this stable mode of regulation over time, a second group of targets consists exclusively of TFs (*cfi-1/Arid3a, bnc-1/BNC1-2, mab-9/Tbx20, ceh-44/CUX1-2, nhr-40/nuclear hormone receptor*) that do not require UNC-3 for initiation, but depend on UNC-3 activity for maintenance. Hence, the suite of UNC-3 targets in cholinergic MNs is modified across different life stages, a phenomenon we term 'temporal modularity' in terminal selector function. To provide mechanistic insights, we focused on the second group of targets and identified a molecular mechanism for their *unc-3*-independent initiation that relies on Hox proteins. Importantly, preventing UNC-3's ability to selectively maintain expression of a single TF (*cfi-1/Arid3a*) from the second target group led to locomotion defects, indicating minimal disruption of temporal modularity affects animal behavior. Lastly, we provide evidence for temporal modularity in the function of UNC-30/PITX, the terminal selector of GABAergic MN identity in *C. elegans* (*Eastman et al., 1999*; *Jin et al., 1994*). Because terminal selectors have been identified in both invertebrate and vertebrate nervous systems (*Deneris and Hobert, 2014*; *Hobert and Kratsios, 2019*), we hypothesize that temporal modularity in their function may be a general mechanism for the establishment and maintenance of neuronal identity.

## Results

### Identifying the global targets of UNC-3 via ChIP-Seq

To identify in an unbiased manner putative UNC-3 target genes, we employed chromatin immunoprecipitation followed by DNA sequencing (ChIP-Seq). We used a reporter strain with in-frame GFP sequences inserted immediately upstream of the stop codon of the endogenous *unc-3* gene (*Figure 1A*). Expression of UNC-3::GFP fusion protein was observed in the nucleus of 53 cholinergic MNs (SAB subtype = 3 neurons, DA = 9, DB = 7, VA = 12, VB = 11, AS = 11) and 19 other neurons known to express *unc-3* (*Figure 1A*), indicating that this reporter faithfully recapitulates the endogenous *unc-3* expression pattern (*Pereira et al., 2015*; *Prasad et al., 1998*). Insertion of GFP does not detectably alter the function of UNC-3 since expression of known UNC-3 targets (*cho-1/ChT, unc-17/VAChT, ace-2/AChE*) is unaffected in *unc-3::gfp* animals (*Figure 1B*). Unlike *unc-3* null mutants, *unc-3::gfp* animals do not display locomotion defects. We therefore performed ChIP-Seq on *unc-3::gfp* animals at larval stage 2 (L2) because all *unc-3* expressing neurons have been generated by this stage.

Our ChIP-Seq dataset revealed strong enrichment of UNC-3 binding in the genome by identifying a total of 6892 unique binding peaks (q-value cutoff: 0.05) (*Figure 1C* and *Figure 1—figure supplement 1*). The majority of UNC-3 binding peaks (91.95%) are predominantly located between 0 and 3 kb upstream of a transcription start site, whereas only a small fraction is found in introns (2.05%) or downstream of a gene (0.99%), suggesting UNC-3 chiefly acts at promoters and enhancers to regulate gene expression (*Figure 1D and F*). Through de novo motif discovery analysis (see Materials and methods), we identified a 12 bp pseudo-palindromic sequence overrepresented in the UNC-3 binding peaks (*Figure 1E*), which highly resembles the binding site of UNC-3 vertebrate orthologs (*Treiber et al., 2010a*; *Treiber et al., 2010b*; *Wang and Reed, 1993*; *Wang et al., 1997*). To test the quality of our ChIP-Seq results, we sought to determine whether UNC-3 binding peaks are present in the *cis*-regulatory region of all previously described UNC-3 targets in cholinergic MNs, because these neurons constitute the majority of *unc-3*-expressing cells (53 out of 72 cells). Previous studies identified 10 terminal identity genes as putative direct UNC-3 targets and 43 terminal identity genes whose expression is affected by genetic removal of *unc-3* (*Kratsios et al., 2015*; *Kratsios et al., 2011*). In the current study, we found UNC-3 binding peaks in 9 out of 10 (90%) direct UNC-3 targets and 38 of the 43 (88.37%) downstream targets of UNC-3, indicating high quality in the ChIP-Seq results (*Figure 2A* and *Figure 1—figure supplement 1*, *Supplementary file 1*). Moreover, ChIP-Seq for UNC-3 appears highly sensitive, as it identified peaks in *unc-3*-dependent genes expressed in a limited number of neurons (e.g., *glr-4/GluR* is expressed in 4 out of 72 *unc-3+* neurons) (*Figure 2A*). In conclusion, our ChIP-Seq experiment generated a comprehensive map of UNC-3 binding in the *C. elegans* genome and provided biochemical evidence to the hypothesis that UNC-3 binds to the *cis*-regulatory region of multiple terminal identity genes, consolidating UNC-3's function as a terminal selector of cholinergic MN identity.

Our bioinformatic analysis of the UNC-3 binding peaks revealed 3502 protein-coding genes as putative UNC-3 targets (see Materials and methods). To classify these new targets, we performed

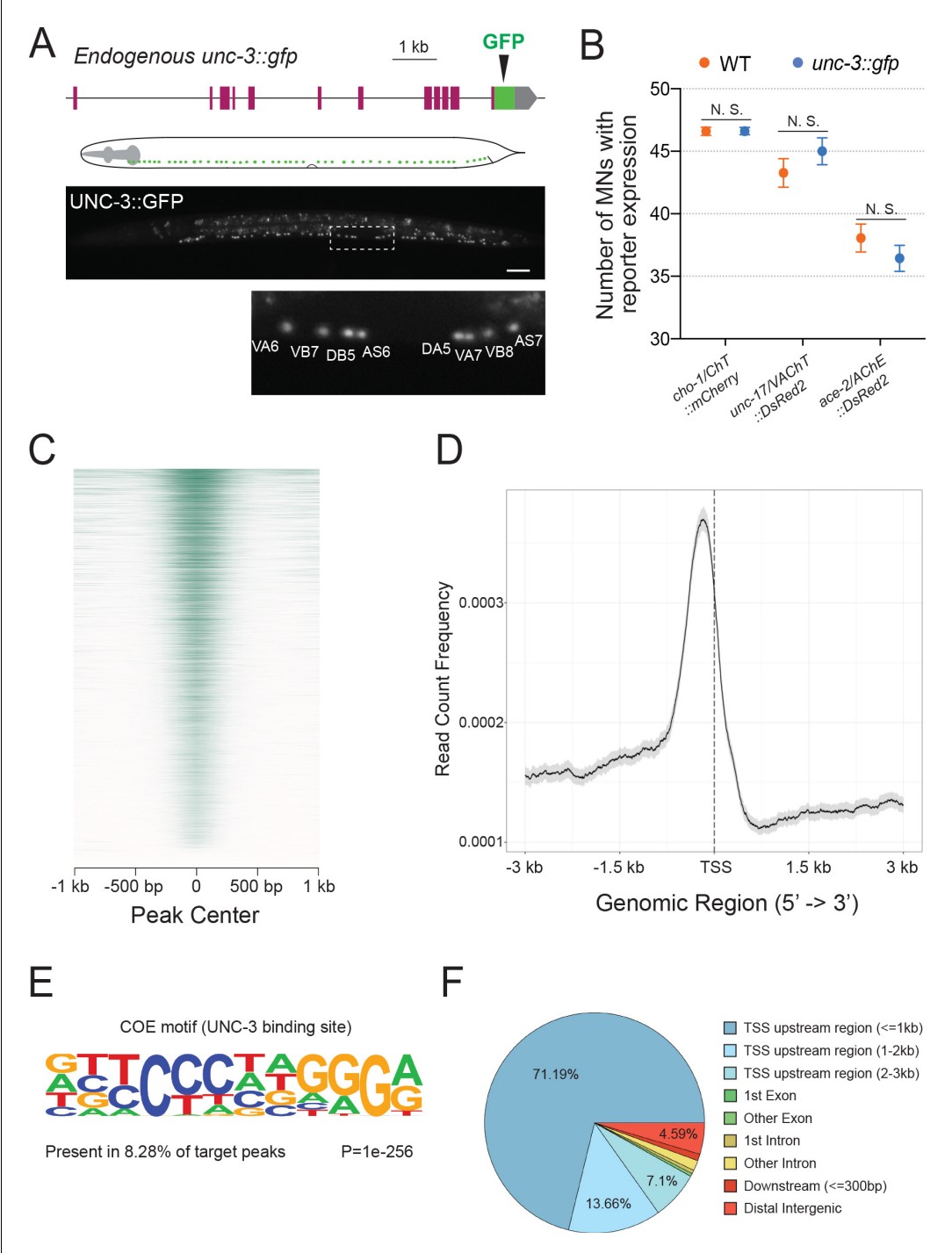

**Figure 1.** Mapping UNC-3 binding genome-wide with ChIP-Seq. (**A**) Diagram illustrating the endogenous reporter allele of UNC-3. GFP is inserted immediately upstream of *unc-3*'s stop codon. Below, a representative image at L2 stage showing expression of UNC-3::GFP fusion protein in cholinergic MN nuclei. Region highlighted in dashed box is enlarged. Scale bar, 20 μm. (**B**) Quantification of terminal identity gene markers that report expression of known UNC-3 targets (*cho-1/ChT, unc-17/VAChT, ace-2/AChE*) in WT and *ot839 [unc-3::gfp]* animals at the L4 stage (N = 15). N. S.: not significant. (**C**) Heatmap of UNC-3 ChIP-Seq signal around 1.0 kb of the center of the binding peak. (**D**) Summary plot of UNC-3 ChIP-Seq signal with a 95% confidence interval (gray area) around 3.0 kb of the TSS. The average signal peak is detected at ~200 bp upstream of the TSS. (**E**) de novo motif discovery analysis of 6,892 UNC-3 binding peaks identifies a 12 bp-long UNC-3 binding motif. (**F**) Pie chart summarizes genomic distribution of UNC-3 ChIP-Seq signal.

The online version of this article includes the following figure supplement(s) for figure 1:

*Figure 1 continued on next page*

*Figure 1 continued*

**Figure supplement 1.** UNC-3 ChIP-Seq results yield genome-wide enrichment of UNC-3.

gene ontology (GO) analysis focused on protein classes using PANTHER (*Mi et al., 2013*). This analysis revealed three broad categories (*Figure 2B*, *Supplementary file 2*). First, there is a preponderance (42.18% of the total number of UNC-3 targets classified by PANTHER) of terminal identity genes (e.g., 114 transporter proteins, 111 receptors and trans-membrane proteins, 37 signaling molecules, 11 cell adhesion molecules), suggesting that UNC-3 broadly affects multiple features of neuronal terminal identity. The second overrepresented category (24.07% of UNC-3 targets) contains a large number of proteins involved in the control of gene expression, such as 239 nucleic acid-binding proteins (16.77%) and 104 TFs (7.3%), highlighting the possibility of an extensive network of gene regulatory factors downstream of UNC-3. The third category (24.14%) consists of genes coding for various types of enzymes (e.g., hydrolases, ligases, oxidoreductases), suggesting a new role for UNC-3 in neuronal metabolic pathways. Together, this analysis unravels the breadth of biological processes potentially under the direct control of UNC-3.

The ChIP-Seq experiment was performed on endogenously tagged UNC-3, which is expressed in 53 cholinergic MNs of the nerve cord and 19 other neurons located in the *C. elegans* head and tail (*Pereira et al., 2015*). Since MNs are the majority (53 cells) of *unc-3*-expressing cells (72 in total), a significant portion of the UNC-3-bound genes may be expressed in MNs. To test this, we used available single-cell RNA-Seq data (CeNGEN project: www.cengen.org) that identified 576 transcripts specifically enriched in cholinergic MNs (*Taylor et al., 2019*). We found that 52.95% of these MN-expressed genes are bound by UNC-3 and fall in the aforementioned gene categories (*Supplementary file 3*), thereby constituting putative UNC-3 targets in cholinergic MNs.

## *Cis*-regulatory analysis reveals novel TFs as direct UNC-3 targets in MNs

Our ChIP-Seq results provide an opportunity to reveal new roles for UNC-3, beyond the direct control of terminal identity genes. To this end, we focused on the 104 TFs identified by ChIP-Seq as putative UNC-3 targets (*Figure 2B–C*). To functionally test whether the UNC-3 bound DNA elements upstream of these TF-encoding genes carry information critical for gene expression, we carried out a *cis*-regulatory analysis. We isolated and fused to RFP elements located upstream of 16 randomly selected TFs of different families (e.g., homeobox, nuclear hormone receptors, Zn finger) (*Figure 2C*, *Table 1*). We generated transgenic reporter animals and found that 10 of these TF reporters were sufficient to drive RFP expression in ventral cord MNs (*Table 1*). Next, we identified 9 TFs (*nhr-1, nhr-40, mab-9, ztf-26, ceh-44, zfh-2, cfi-1, bnc-1, nhr-19*) with expression in cholinergic MNs. Some of those are also expressed in the *unc-3*-negative GABAergic MNs of the nerve cord (*Table 1*). Interestingly, one reporter (*nhr-49*) is exclusively expressed in GABAergic MNs. Expression of five TFs (*nhr-1, nhr-19, nhr-49, zfh-2, ztf-26*) in ventral cord MNs has not been previously described. The remaining 5 TFs (*cfi-1/Arid3a, bnc-1/Bnc1/2, mab-9/Tbx20, nhr-40, ceh-44/Cux1*) are known to be expressed in subsets of *unc-3*-positive MNs (*Kerk et al., 2017*; *Pocock et al., 2008*; *Brozová et al., 2006*), but our analysis revealed *cis*-regulatory elements sufficient for their MN expression. Next, we tested for *unc-3* dependency at the L4 stage, and found that 8 of the 9 TF reporters (*nhr-1, nhr-40, zfh-2, ztf-26, ceh-44, cfi-1, bnc-1, mab-9*) with expression in cholinergic MNs are positively regulated by *unc-3*; their expression significantly decreased in *unc-3* mutants (*Table 1*, *Figure 3—figure supplement 1*). On the other hand, *nhr-49*, which is normally expressed in GABAergic MNs, is negatively regulated by *unc-3*; ectopic expression was observed in cholinergic MNs of *unc-3* mutants (*Table 1*, *Figure 3—figure supplement 1*). Together, this *cis*-regulatory analysis revealed novel TFs as direct UNC-3 targets in cholinergic MNs. Besides its known role as an activator of terminal identity genes (*Kratsios et al., 2015*; *Kratsios et al., 2011*), these findings suggest UNC-3 can also act directly to either activate or repress expression of multiple TF-encoding genes. Alternatively, UNC-3 binding could facilitate recruitment of other factors that function either as activators or repressors. Altogether, these data uncover an extensive gene regulatory network downstream of UNC-3.

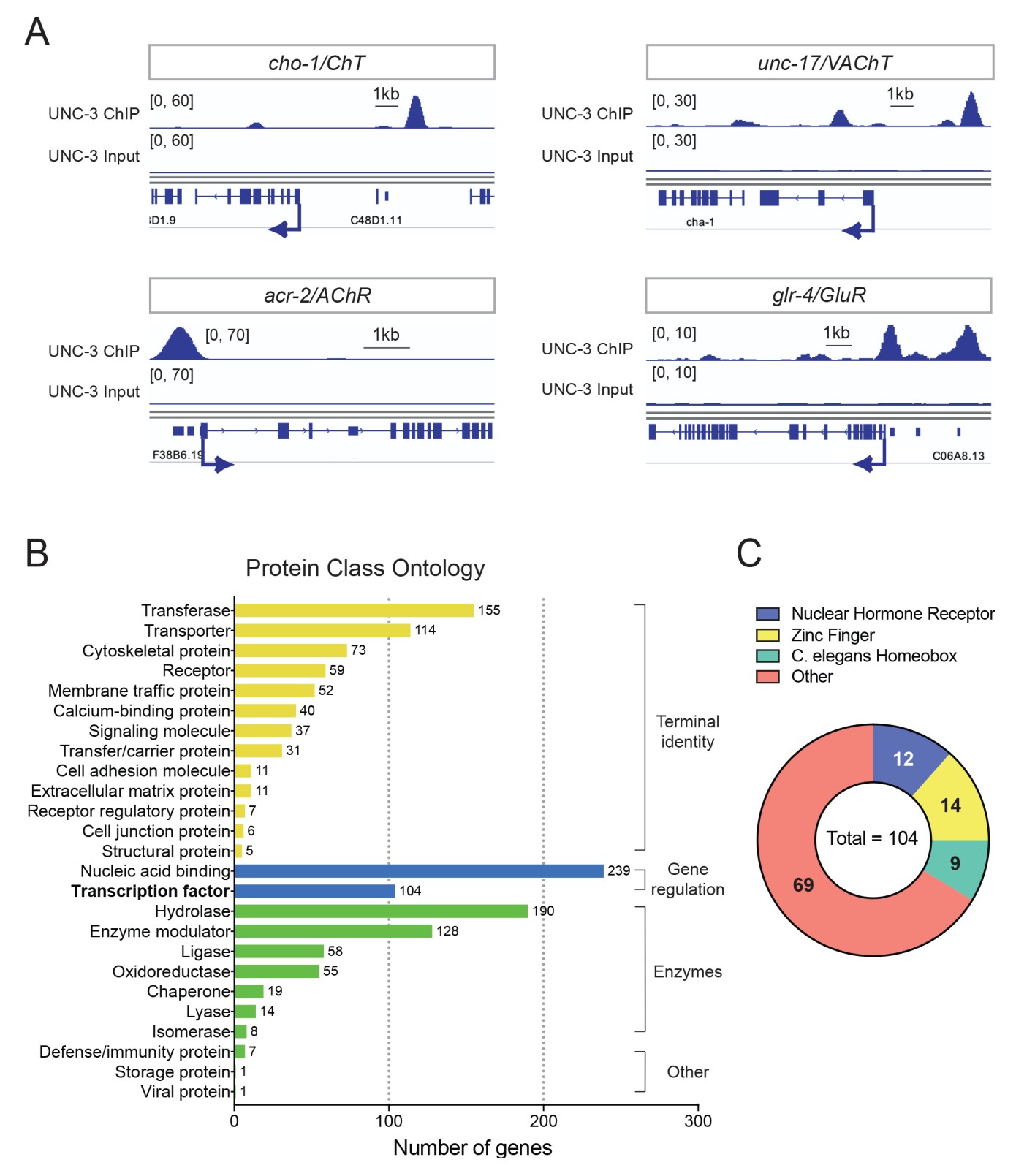

**Figure 2.** Global analyses of UNC-3 ChIP-Seq data. (**A**) Snapshots of UNC-3 ChIP-Seq and input (negative control) signals at the *cis*-regulatory regions of known UNC-3 targets (*cho-1/ChT, unc-17/VAChT, acr-2/AChR, glr-4/GluR*). (**B**) Graph summarizing protein class ontology analysis of putative target genes of UNC-3 identified by ChIP-Seq. Out of the 3502 protein-coding UNC-3 targets, 1425 encode for proteins with known protein class terms and

*Figure 2 continued on next page*

*Figure 2 continued*

these were the ones considered by PANTHER. This analysis classifies UNC-3 targets into three broad categories: terminal identity genes, gene expression regulators, and enzymes. (**C**) Pie chart breaking down TF families that show UNC-3 binding.

**Table 1.** Summary of *cis*-regulatory analysis to identify novel transcription factors controlled by UNC-3.

Transgenic reporter (RFP) animals for each TF were generated and examined for neuronal expression. GFP markers for cholinergic and GABAergic MNs were used to define TF reporter expression. TF reporters expressed in MNs were further tested for UNC-3 dependency. Eighteen reporters were generated that correspond to 16 TFs (Two reporters were generated for *nhr-1* and *nhr-40* because two distinct UNC-3 ChIP-Seq peaks were found in the *cis*-regulatory region of these genes). Not applicable (N/A).

| Number of reporters | Gene | TF family | DNA region included relative to ATG | unc-3 dependence | RFP expression | Expression in cholinergic MNs | Expression in GABAergic MNs | # of lines |
|---|---|---|---|---|---|---|---|---|
| 1 | *nhr-1* (second peak) | Nuclear hormone receptor | −838 bp to +1,535 bp | Positively regulated | VNC, Head and Tail Neurons | YES | YES | 2 |
| 2 | *nhr-40* (second peak) | Nuclear hormone receptor | +4,356 bp to +5,525 bp | Positively regulated | VNC, Head and Tail Neurons | YES | YES | 2 |
| 3 | *mab-9* | T-box | −5,561 bp to −3,773 bp | Positively regulated | VNC, Head and Tail Neurons | YES | YES | 2 |
| 4 | *ztf-26* | Zinc finger | +910 bp to 3,394 bp | Positively regulated | VNC and Head Neurons | YES | NO | 2 |
| 5 | *ceh-44* | *C. elegans* homeobox | +1,605 bp to 3,111 bp | Positively regulated | VNC, Head and Tail Neurons | YES | Not determined | 2 |
| 6 | *zfh-2* | Zinc finger | +12,048 bp to +13,549 bp | Positively regulated | VNC, Head and Tail Neurons | YES | NO | 2 |
| 7 | *cfi-1* | AT-rich interaction domain | −13,824 bp to −11,329 bp | Positively regulated | VNC, Head and Tail Neurons | YES | YES | 2 |
| 8 | *bnc-1* | Zinc finger | −1,800 bp to −1 bp | Positively regulated | VNC and Tail Neurons | YES | NO | 2 |
| 9 | *nhr-19* | Nuclear hormone receptor | +1,261 bp to +2,294 bp | NO | VNC, Head and Tail Neurons | YES | YES | 2 |
| 10 | *nhr-40* (first peak) | Nuclear hormone receptor | +919 bp to +1,839 bp | NO | VNC, Head and Tail Neurons | YES | YES | 2 |
| 11 | *nhr-49* | Nuclear hormone receptor | −750 bp to 75 bp | Negatively regulated | VNC, Head and Tail Neurons | NO | YES | 2 |
| 12 | *nhr-1* (first peak) | Nuclear hormone receptor | −7,452 bp to −5,871 bp | Not determined | Head and Tail Neurons | N/A | N/A | 2 |
| 13 | *ccch-3* | Zinc finger | −385 bp to −1 bp | Not determined | Head Neurons | N/A | N/A | 2 |
| 14 | *ztf-17* | Zinc finger | −1,031 bp to +3 bp | Not determined | Head Neurons, Muscles, and Intestine | N/A | N/A | 2 |
| 15 | *nhr-47* | Nuclear hormone receptor | −675 bp to +171 bp | N/A | No RFP expression | N/A | N/A | 2 |
| 16 | *unc-55* | Nuclear receptor | −2,105 bp to −1,327 bp | N/A | No RFP expression | N/A | N/A | 2 |
| 17 | *ztf-13* | Zinc finger | −1,392 bp to −1 bp | N/A | No RFP expression | N/A | N/A | 2 |
| 18 | *ztf-14* | Zinc finger | −2,087 bp to −531 bp | N/A | No RFP expression | N/A | N/A | 2 |

## Temporal modularity of UNC-3 function in cholinergic MNs

Terminal selectors are continuously expressed, from development through adulthood, in specific neuron types. However, it remains unclear whether – in the same neuron type – a terminal selector controls an identical suite of targets across different life stages, or the suite of targets can change over time. The case of UNC-3 offers an opportunity to address this issue because its direct targets (terminal identity genes and newly identified TFs [*Table 1*]) are continuously expressed in cholinergic MNs. However, *unc-3* dependency of terminal identity genes was mostly tested at a single developmental stage, the last larval stage (L4) (*Kratsios et al., 2011*). We therefore performed a longitudinal analysis to determine whether target gene dependency on *unc-3* remains stable or changes at different life stages.

First, we tested four terminal identity genes (*acr-2/AChR, unc-129/TGFbeta, glr-4/GluR, unc-17/VAChT*) at larval (L2, L4) and adult (day 1) stages and found that their MN expression, at every stage, critically depends on UNC-3 (*Figure 3A*). Because this analysis relied on animals lacking *unc-3* gene activity since early development (a null allele was used), whether UNC-3 is continuously required to maintain expression of these genes remained unclear. We therefore employed the auxin-inducible degradation (AID) system to deplete the endogenous UNC-3 protein in cholinergic MNs at late larval and young adult stages (*Zhang et al., 2015*). Compared to controls, we found a significant decrease in the number of adult MNs expressing *acr-2/AChR, glr-4/GluR*, and *unc-17/VAChT* in animals treated with auxin (*Figure 3—figure supplement 2*). These findings indicate UNC-3 is required to initiate and maintain the expression of terminal identity genes, consolidating its role as a terminal selector of cholinergic MN fate.

However, a different picture emerged after testing the 8 TF reporters that are positively regulated by UNC-3 (*Table 1*). Similar to terminal identity genes, the expression of 3 TFs (*zfh-2/Zfhx3, ztf-26, nhr-1*) critically depends on UNC-3 at every stage (L2, L4, adult) (*Figure 3—figure supplement 1*), suggesting UNC-3 controls initiation and maintenance of their expression. In striking contrast, the early expression (L2 stage) of five TFs (*cfi-1, bnc-1, mab-9, nhr-40, ceh-44*) does not require UNC-3 (*Figure 3B*). However, maintenance of their expression during late larval and/or adult stages does depend on UNC-3 (*Figure 3B*). Hence, this longitudinal analysis revealed two groups of targets with distinct requirements for UNC-3 at different life stages. One group consists of terminal identity genes (*acr-2/AChR, unc-129/TGFbeta, glr-4/GluR, unc-17/VAChT*) and TFs (*zfh-2/Zfhx3, ztf-26, nhr-1*) that require UNC-3 for both initiation and maintenance of expression ('initiation and maintenance' module, *Figure 3A,C*). The second group consists exclusively of TFs (*cfi-1/Arid3a, bnc-1/Bnc1, mab-9/Tbx20, ceh-44/Cux1, nhr-40*) that depend on UNC-3 for maintenance, but not initiation ('maintenance-only' module, *Figure 3B–C*).

Collectively, these findings suggest that, in cholinergic MNs, the suite of UNC-3 target genes can partially change at different life stages. In all larval and adult stages examined, UNC-3 is required for the continuous expression of one set of genes ('initiation and maintenance' module). However, only in late larvae and adults, UNC-3 is required to maintain expression of another set of genes ('maintenance-only' module). To describe this phenomenon, we use the term 'temporal modularity in UNC-3 function' given that the function of a TF in a specific cell type and at a particular life stage is determined by the suite of targets it controls in that cell type and at that stage (*Figure 3C*). In the following sections, we hone in on a single target (*cfi-1/Arid3a*) from the 'maintenance-only' module, aiming to dissect the molecular mechanisms underlying the temporal modularity of UNC-3 function in cholinergic MNs.

## A distal enhancer is necessary for initiation and maintenance of *cfi-1/Arid3a* expression in MNs

Our *cis*-regulatory analysis suggests that maintenance, but not initiation, of *cfi-1* expression depends on UNC-3 (*Figure 3B*). We therefore hypothesized that the sole UNC-3 binding peak on the *cfi-1* locus (located ~12 kb upstream) demarcates an enhancer element selectively required for maintenance (*Figures 3B* and *4A*). If this were to be the case, then it would be logical to assume that a separate *cis*-regulatory element would control *cfi-1* initiation in MNs. To test this assumption, we conducted an unbiased *cis*-regulatory analysis *in vivo* by generating a series of 12 transgenic reporter (GFP or RFP) animals, with each reporter carrying small and contiguous DNA fragments spanning a ~15 kb region (*Figure 4A*). Surprisingly, this analysis did not reveal a separate initiation

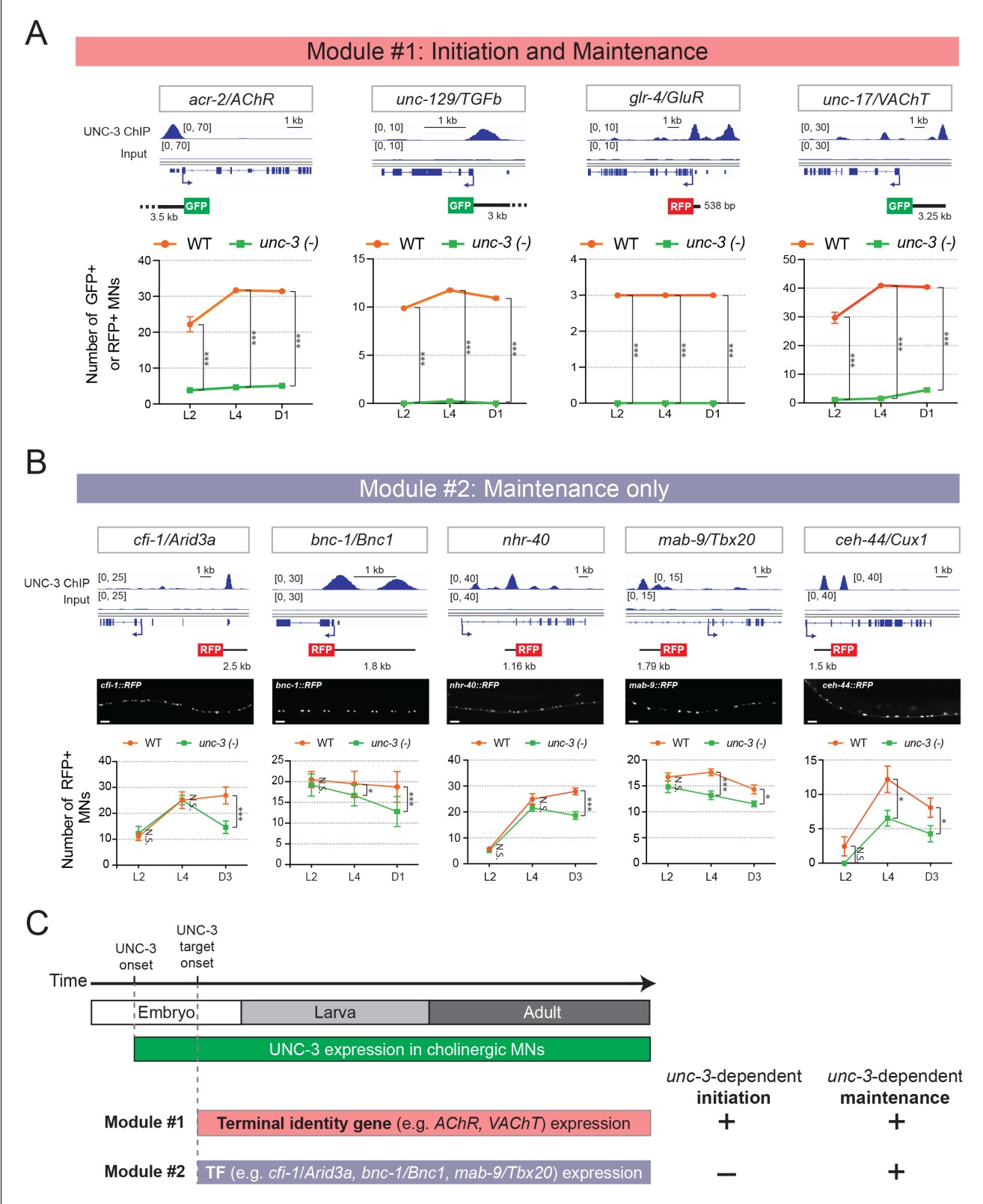

**Figure 3.** Terminal identity genes and transcription factors display distinct temporal requirements for UNC-3. (**A**) Top: snapshots of UNC-3 ChIP-Seq and input (negative control) signals at the *cis*-regulatory regions of four cholinergic terminal identity genes (*acr-2/AChR, unc-129/TFGb, glr-4/GluR,* and *unc-17/VAChT*). The length of DNA elements included in each reporter is shown. Bottom: quantification of terminal identity gene reporters in WT and *unc-3 (n3435)* animals at three different developmental stages – L2, L4, and day 1 adults (N ≥ 12). UNC-3 is required for both initiation and

*Figure 3 continued on next page*

Figure 3 continued

maintenance of all four terminal identity genes. \*\*\*p<0.001. (B) Top: snapshots of UNC-3 ChIP-Seq and input (negative control) signals at the *cis*-regulatory regions of 5 transcription factors (*cfi-1/Arid3a, bnc-1/Bnc1, nhr-40, mab-9/Tbx20,* and *ceh-44/Cux1*). The length of DNA elements included in each reporter is shown. Middle: representative images of WT L4 animals showing expression of the transgenic reporters in MNs. Scale bar, 20 μm. Bottom: quantification of transcription factor reporters in WT and *unc-3 (n3435)* animals at three different developmental stages – L2, L4, and young adults (day 1 or day 3) (N ≥ 12). UNC-3 is required for maintenance, but not initiation of the 5 TFs. N.S.: not significant, \*p<0.05, \*\*\*p<0.001. (C) Schematic summarizing the phenomenon of temporal modularity in UNC-3 function. The first module consists of terminal identity genes and TFs (*Figure 3—figure supplement 2*), which require UNC-3 for both initiation and maintenance of gene expression. The second module consists exclusively of TFs that require UNC-3 only for maintenance.

The online version of this article includes the following figure supplement(s) for figure 3:

**Figure supplement 1.** UNC-3 directly controls the expression of several TF reporters in MNs.

**Figure supplement 2.** UNC-3 is required to maintain the expression of *glr-4/GluR, unc-17/VAChT,* and *acr-2/AChR* in cholinergic motor neurons.

element. Instead, it showed that the same 2.5 kb distal element (reporter #7) that drives RFP expression in subsets of *unc-3*-expressing cholinergic MNs (DA, DB, VA, VB subtypes) at larval and adult stages (*Figure 3B*), is also sufficient for embryonic (3-fold stage) MN expression (*Figure 4A–C*). In addition, this 2.5 kb element also showed expression at all these stages in *unc-3*-negative neurons of the nerve cord, namely the GABAergic (DD, VD subtypes) MNs (*Figure 4A,C*), which will be discussed later in Results. We conclude that this enhancer element is sufficient for initiation and maintenance of *cfi-1* reporter expression in nerve cord MNs.

To test the necessity of this element, we first generated via CRISPR/Cas9 an endogenous mNeon-Green (mNG) reporter allele for *cfi-1*, which also carries an auxin-inducible degron (AID) tag (*mNG:: AID::cfi-1*), enabling inducible depletion of CFI-1 (depletion experiments are described later in Results). Animals carrying the *mNG::AID::cfi-1* allele do not show any developmental phenotypes, suggesting that the mNG::AID tag does not detectably alter *cfi-1* activity. This reporter showed expression in subsets of *unc-3*-expressing MNs (DA, DB, VA, VB subtypes), GABAergic nerve cord MNs (DD, VD subtypes), tail and head neurons, as well as head muscle (*Figure 4A–C*), a pattern consistent with previous studies describing *cfi-1* expression (*Shaham and Bargmann, 2002*; *Kerk et al., 2017*). We determined the onset of the endogenous *cfi-1* reporter (*mNG::AID::cfi-1)* in MNs to be at the 3-fold embryonic stage, coinciding with the onset of transgenic reporters (#7 and #8) containing the distal enhancer (*Figure 4B*). Next, we employed CRISPR/Cas9 genome editing and deleted 769 bp that constitute the core of the UNC-3 binding peak (located ~12 kb upstream) in the context of the endogenous *cfi-1* reporter (*mNG::AID::cfi-1$^{\Delta\ enhancer\ (769\ bp)}$*). We found that *mNG::AID::cfi-1* expression is selectively eliminated in cholinergic (DA, DB, VA, VB) and GABAergic (DD, VD) nerve cord MNs at all life stages examined (3-fold embryo, L4, Day 1 adult) (*Figure 4A–B*, quantification of cholinergic MNs shown in *Figure 4D*).

We conclude that a distal 2.5 kb enhancer (located ~12 kb upstream of *cfi-1*) is sufficient for *cfi-1* expression in nerve cord MNs. Genome editing suggests that a 769 bp sequence within this 2.5 kb enhancer is required for both initiation and maintenance of *cfi-1* in nerve cord MNs (*Figure 4D*). In the ensuing sections, we test the hypothesis that this enhancer integrates UNC-3 input for *cfi-1* maintenance in cholinergic MNs, as well as UNC-3-independent input for *cfi-1* initiation in these neurons.

## UNC-3 maintains *cfi-1* expression in cholinergic MNs via direct activation of the distal enhancer

The binding of UNC-3 to the distal enhancer strongly suggests UNC-3 acts directly to maintain *cfi-1* expression in cholinergic MNs (*Figure 4A*). However, the UNC-3 peak is spread across several hundred base pairs due to the inherently low ChIP-Seq resolution. Hence, the precise DNA sequences recognized by UNC-3 remained unknown. Through bioinformatic analysis (see Materials and methods), we identified eight putative UNC-3 binding sites (COE motifs) within the 769 bp distal enhancer (*Figure 4E*). Using CRISPR/Cas9 technology, we simultaneously mutated all eight motifs in the context of the endogenous *cfi-1* reporter allele (*mNG::AID::cfi-1$^{8\ COE\ motifs\ mut}$*) by substituting nucleotides known to be critical for DNA binding of UNC-3 orthologs (*Treiber et al., 2010a*; *Wang et al., 1993*; *Figure 4E*). During the L2 stage, expression of *mNG* in MNs is not affected in *mNG::AID::cfi-1 $^{8\ COE\ motifs\ mut}$* animals, indicating early *cfi-1* expression occurs normally (*Figure 4F*).

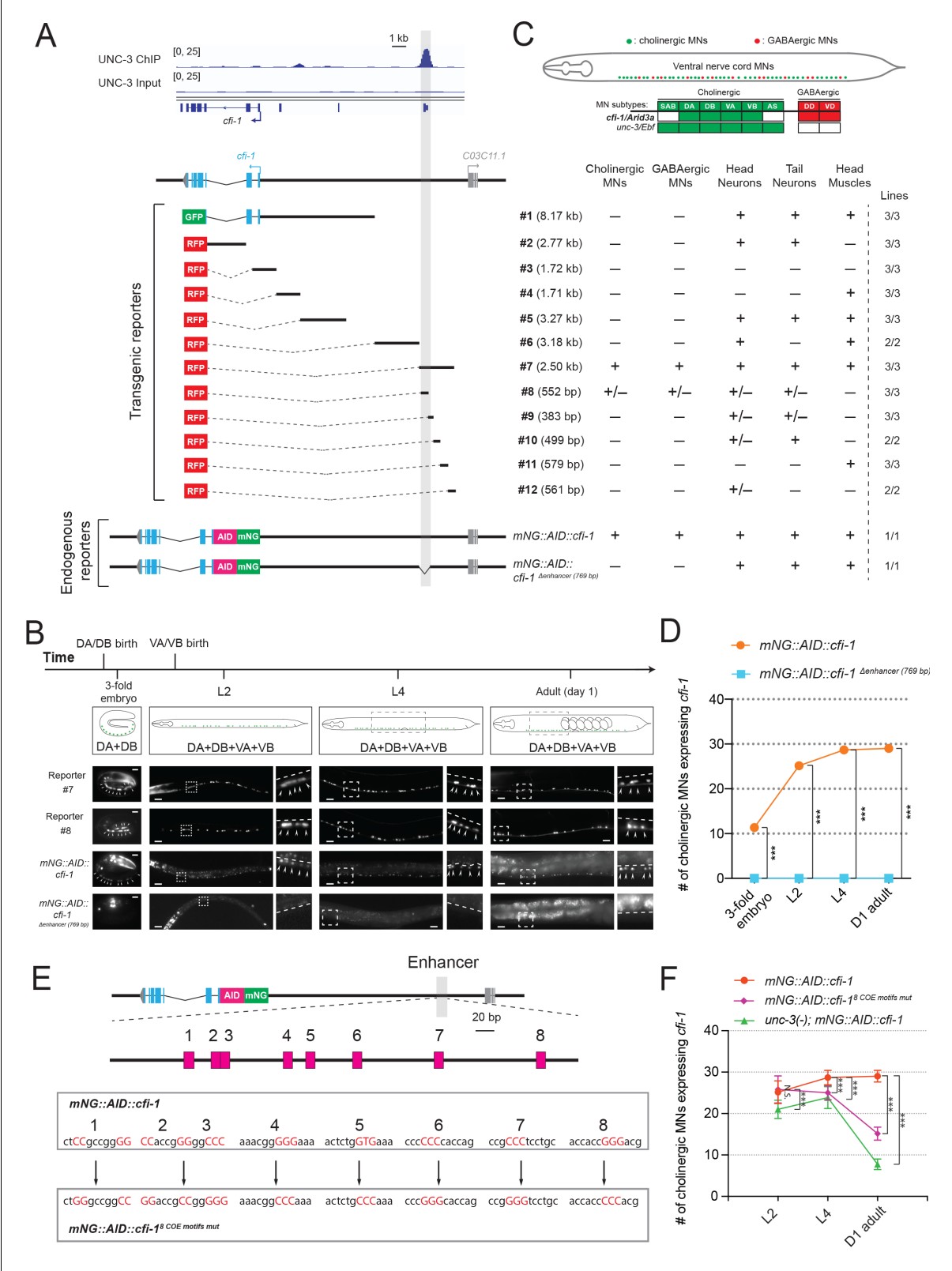

**Figure 4.** UNC-3 acts through a distal enhancer to maintain *cfi-1* expression in cholinergic motor neurons. (**A**) Top: Snapshots of UNC-3 ChIP-Seq and input (negative control) signals at the *cis*-regulatory region of *cfi-1*. The gray bar highlights an UNC-3 binding peak located ~12 kb upstream of the TSS of *cfi-1* (−11,391 bp to −12,146 bp). Bottom: schematic showing the strategy of constructing *cfi-1* reporters. Twelve transcriptional fusion reporters ([−1 bp to −8,170 bp], #2 [993 bp to 3,764 bp], #3 [547 bp to −1,173 bp], #4 [-1,164 bp to −2,875 bp], #5 [−2,865 bp to −6,141 bp], #6 [-8,162 bp to −11,346

*Figure 4 continued on next page*

Figure 4 continued

bp], #7 [−11,329 bp to −13,824 bp], #8 [−11,329 bp to −11,881 bp], #9 [−11,851 bp to −12,234 bp], #10 [−12,223 bp to −12,722 bp], #11 [−12,705 bp to −13,284 bp], and #12 [-13,263 bp to −13,824 bp]) carry *cis*-regulatory regions fused to fluorescent reporters (GFP or RFP). The endogenous reporter alleles (*mNG::AID::cfi-1* and *mNG::AID::cfi-1* <sup>Δenhancer (769 bp)</sup>) have an in-frame fluorescent protein mNeonGreen (mNG) insertion immediately after the ATG of *cfi-1*. The enhancer KO allele *mNG::AID::cfi-1* <sup>Δenhancer (769 bp)</sup> carries a 769 bp deletion (−11,329 bp to −12,097 bp). Table on the right summarizes the expression pattern of each reporter allele at L4 stage. N ≥ 12. +: reporter expressed, –: reporter not expressed, +/–: reporter partially expressed in the respective neurons. Number of independent transgenic lines tested for each reporter is shown on the right. (B) Representative images showing the expression of reporter #7, reporter #8, *mNG::AID::cfi-1*, and *mNG::AID::cfi-1* <sup>Δenhancer (769 bp)</sup> at specific life stages. Areas highlighted in dashed boxes are enlarged and presented on the right side of each picture. The onset of *cfi-1* expression occurs at the 3-fold embryonic stage. mNG+ MNs are annotated with arrowheads. Scale bars, 5 μm (3-fold embryos); 20 μm (larvae and adults). (C) Schematic summarizing the expression pattern of *cfi-1* and *unc-3* in nerve cord MNs. (D) Quantification of the number of cholinergic MNs expressing endogenous *cfi-1* (*mNG::AID::cfi-1*) in WT and animals carrying the enhancer deletion (*mNG::AID::cfi-1* <sup>Δenhancer (769 bp)</sup>). Deletion of the enhancer element located ~12 kb upstream of the TSS of *cfi-1* completely abolishes *cfi-1* expression in MNs at all tested stages. A red fluorescent marker (*ttr-39::mCherry*) for GABAergic MNs was used to exclude these neurons from the quantification. Cholinergic MNs expressing *cfi-1* were positive for mNG and negative for mCherry. (E) Bioinformatic analysis predicted 8 UNC-3 binding sites (COE motifs, shown as pink boxes) in the *cfi-1* enhancer region, which displays UNC-3 binding (−11,391 bp to −12,146 bp). Using CRISPR/Cas9, these eight motifs were mutated by substituting duplets or triplets of nucleotides as shown below, thereby generating the strain *cfi-1 (syb1856 [mNG::AID::cfi-1*<sup>8 COE motifs mut</sup>]). (F) Quantification of the number of cholinergic MNs expressing the endogenous *cfi-1* reporter (*mNG::AID::cfi-1*) in WT and *unc-3 (n3435)* animals, as well as in animals with mutated COE motifs (*mNG::AID::cfi-1*<sup>8 COE motifs mut</sup>) at L2, L4, and day 1 adult stages (N ≥ 12). N.S.: not significant, ***p<0.001.

The online version of this article includes the following figure supplement(s) for figure 4:

**Figure supplement 1.** CFI-1 does not auto-regulate its expression in motor neurons.

Intriguingly, *mNG* expression is significantly down-regulated in cholinergic MNs at later larval (L4) and adult (day 1) stages, resembling the phenotype of *unc-3* null mutants (*Figure 4F*). We conclude that UNC-3 binds to the distal enhancer and directly acts through one or more of these 8 COE motifs to maintain *cfi-1* expression in cholinergic MNs.

Previous studies in the nervous system have shown that a TF can maintain its own expression via transcriptional activation either by itself (positive auto-regulation), or in partnership with other TFs (*Leyva-Díaz and Hobert, 2019*; *Scott et al., 2005*; *Xue et al., 1992*). We found though that *cfi-1* does not auto-regulate and UNC-3 binding at the distal enhancer occurs normally in *cfi-1* null mutants (*Figure 4—figure supplement 1*), excluding a potential involvement of CFI-1 in its own maintenance.

## LIN-39 (Scr/Dfd/HOX4-5) and MAB-5 (Antp/HOX6-8) control *cfi-1* expression in cholinergic MNs through the same distal enhancer

If UNC-3 exerts a maintenance role, what are the factors that initiate *cfi-1* expression in MNs? Previous work showed that two Hox proteins, LIN-39 (Scr/Dfd/HOX4-5) and MAB-5 (Antp/HOX6-8), control *cfi-1* expression in MNs (*Kratsios et al., 2017*). However, analysis was performed at the last larval stage (L4) and transgenic *cfi-1* reporter animals were used. Hence, it is unclear whether LIN-39 and MAB-5 are required for initiation of endogenous *cfi-1*.

Because *lin-39; mab-5* double null mutants are viable (*Liu and Fire, 2000*), we performed a longitudinal analysis and found that expression of the endogenous *mNG::AID::cfi-1* reporter in cholinergic MNs is severely affected at embryonic (3-fold), larval (L2, L4) and adult (D1) stages (*Figure 5A–C*). Since onset of *cfi-1* expression occurs at the 3-fold embryonic stage (*Figure 4B*), these results suggest LIN-39 and MAB-5 are required for *cfi-1* initiation. Conversely, initiation of *cfi-1* expression is not affected in a null mutant of *unc-3* (*Figure 5B–C*). Moreover, available ChIP-Seq data from modENCODE (*Boyle et al., 2014*) indicate that LIN-39 and MAB-5 bind to the *cfi-1* distal enhancer (*Figure 5A*). Expression of a 2.5 kb transgenic *cfi-1* reporter (reporter #7) that carries the distal enhancer is significantly affected in *lin-39; mab-5* double mutants at early larval (L2) stages (*Figure 5A,D*). These results strongly suggest that LIN-39 and MAB-5 activate *cfi-1* expression directly. Although the DNA sequence of the MAB-5 binding site is not known, mutation of a single, bioinformatically predicted LIN-39 binding site (wild-type: aaTTGAtg >mutated: aaGGGGtg) within the enhancer led to a decrease in reporter gene expression at L2 (*Figure 5A,E*). This decrease was weaker compared to *lin-39; mab-5* double mutants (*Figure 5C*), likely due to compensation by MAB-5. Indeed, LIN-39 and MAB-5 appear to act synergistically because endogenous *cfi-1* expression

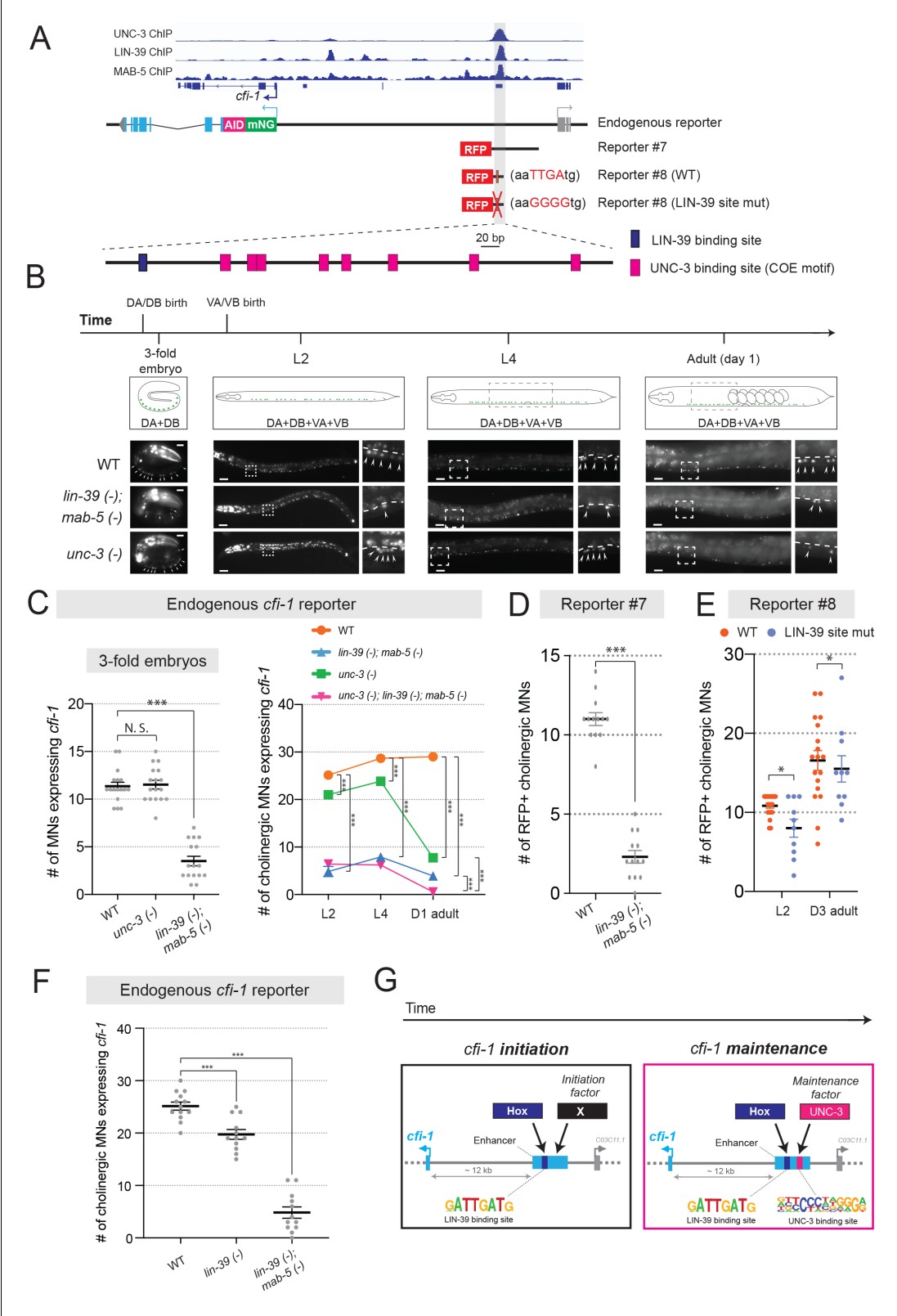

**Figure 5.** UNC-3 and Hox control *cfi-1* expression in cholinergic MNs. (**A**) A snapshot of UNC-3 (L2 stage), LIN-39 (L3 stage), and MAB-5 (L2 stage) ChIP-Seq signals at the *cfi-1* locus. UNC-3, LIN-39, and MAB-5 bind to the same *cfi-1* enhancer (highlighted in gray). Below: Schematics illustrating the reporters used in the rest of the figure. (**B**) Representative images showing the expression of the *mNG::AID::cfi-1* in WT (same images shown in *Figure 4B*), *unc-3 (n3435)*, and *lin-39 (n1760); mab-5 (e1239)* animals during 3-fold embryonic, L2, L4, and day 1 adult stages. *cfi-1* is expressed in four

*Figure 5 continued on next page*

*Figure 5 continued*

cholinergic MN subtypes (DA, DB, VA, and VB). DA and DB are born embryonically, while VA and VB are born post-embryonically. Areas highlighted in dashed boxes are enlarged and presented on the right side of each picture. mNG+ MNs are annotated with arrowheads. Scale bars, 5 μm (3-fold embryos); 20 μm (larvae and adults). (**C**) Quantification of the number of cholinergic MNs expressing the endogenous *cfi-1* reporter (*mNG::AID::cfi-1*) in WT animals, *unc-3 (n3435)* mutants, *lin-39 (n1760); mab-5 (e1239)* double mutants, and *unc-3 (n3435); lin-39 (n1760); mab-5 (e1239)* triple mutants during 3-fold embryonic, L2, L4, and day 1 adult stages (N ≥ 12). N.S.: not significant, \*\*\*p<0.001. A red fluorescent marker (*ttr-39::mCherry*) for GABAergic MNs was used to exclude these neurons from the quantification. Cholinergic MNs expressing *cfi-1* were positive for mNG and negative for mCherry. (**D**) Quantification of the expression of transgenic *cfi-1* reporter #7 in WT and *lin-39 (n1760); mab-5 (e1239)* animals at L2 stage (N = 13). Reporter expression is strongly affected in *lin-39 (-); mab-5 (-)* double mutants. \*\*\*p<0.001. (**E**) Quantification of the WT transgenic reporter #8 and the same reporter with the LIN-39 binding site mutated (point mutations) at larval (**L2**) and adult (**D3**) stages (N ≥ 13). \*p<0.05. (**F**) Quantification of the expression of the *mNG::AID::cfi-1* allele in WT animals, *lin-39 (n1760)* single mutants, and *lin-39 (n1760); mab-5 (e1239)* double mutants at the L2 stage (N ≥ 12). While the number of cholinergic MNs with *cfi-1* expression is mildly decreased in *lin-39* single mutants, more severe effects are observed in double mutants. \*\*\*p<0.001. (**G**) Schematic summarizing the mechanisms underlying initiation and maintenance of *cfi-1* expression in cholinergic MNs. The online version of this article includes the following figure supplement(s) for figure 5:

**Figure supplement 1.** Auxin-inducible depletion of LIN-39 at larval stage 4 (**L4**) does not affect *cfi-1* expression in nerve cord MNs.
**Figure supplement 2.** CFI-1 is required post-embryonically to maintain DA and DB neuronal identities.
**Figure supplement 3.** Hox proteins and UNC-3 control *bnc-1* expression in VA and VB neurons.

(*mNG::AID::cfi-1*) is mildly affected in *lin-39* single mutants, but severely affected in *lin-39; mab-5* double mutants (***Figure 5F***). We conclude that the Hox proteins LIN-39 and MAB-5 are necessary for *cfi-1* initiation in cholinergic MNs (left panel, ***Figure 5G***), and act through the same distal enhancer utilized by UNC-3 to maintain *cfi-1* (right panel, ***Figure 5G***).

Because LIN-39 and MAB-5 are continuously expressed – from embryo through adulthood – in cholinergic MNs (***Feng et al., 2020***), it is likely that these Hox proteins, like UNC-3, are also required for maintenance. To test this, we used the AID system and depleted the endogenous LIN-39 protein at the last larval stage (L4) by using a previously described *lin-39::mNG::AID* allele (***Feng et al., 2020***; ***Zhang et al., 2015***). However, expression of *cfi-1* was unaffected in the adult (***Figure 5—figure supplement 1***). This negative result could be attributed to low and undetectable levels of LIN-39 and/or functional compensation by MAB-5. We therefore used *lin-39; mab-5* double (null) mutants and crossed them to *unc-3* null animals. Of note, Hox genes (*lin-39, mab-5*) and *unc-3* do not cross-regulate their expression (***Kratsios et al., 2017***). If Hox proteins, similar to UNC-3, are required for *cfi-1* maintenance in cholinergic MNs, stronger effects should be present in *unc-3; lin-39; mab-5* triple mutants compared to *unc-3* single mutants. Indeed, we found this to be the case in day 1 adult animals (graph on the right, ***Figure 5C***). Supporting a maintenance role for Hox, mutation of the LIN-39 binding site within the *cfi-1* enhancer (reporter #8) led to a sustained decrease in reporter gene expression from L2 to adult stages (***Figure 5E***).

Together, our findings suggest the Hox proteins LIN-39 and MAB-5 control initiation and maintenance of *cfi-1* in cholinergic MNs via the same distal *cis*-regulatory region (enhancer) utilized by UNC-3 to maintain *cfi-1* (***Figure 5G***). However, this region bears distinct UNC-3 and LIN-39 binding sites.

### *cfi-1/Arid3a* is required post-embryonically to maintain MN subtype identity

Expression of *cfi-1* in cholinergic MNs (DA, DB, VA, VB) is maintained throughout life by Hox and UNC-3 (***Figure 5G***). But why is it important to ensure continuous *cfi-1* expression? Although its function in VA and VB remains unknown, CFI-1 is required during early development to establish the identity of DA and DB subtypes by acting as a transcriptional repressor (***Kerk et al., 2017***). In *cfi-1* null animals, glutamate receptor subunit 4 (*glr-4/GluR*), a terminal identity gene normally activated by UNC-3 in another MN subtype (SAB), becomes ectopically expressed in DA and DB neurons (***Figure 5—figure supplement 2***). In animals lacking *cfi-1* expression specifically in MNs at all stages (*mNG::AID::cfi-1*$^{\Delta\ enhancer\ (769\ bp)}$, ***Figure 4A–B***), we also observed ectopic *glr-4* expression in these neurons (***Figure 5—figure supplement 2***). Hence, early global removal of *cfi-1*, or MN-specific loss of *cfi-1* both lead to DA and DB neurons adopting a mixed identity. However, whether CFI-1 is required post-embryonically to continuously prevent DA and DB from obtaining a mixed identity is not known.

To enable CFI-1 protein depletion selectively at post-embryonic stages, we used the *mNG::AID::cfi-1* reporter allele (*Figure 4A*), which also serves as a conditional allele as it carries the auxin-inducible degron (AID) (*Zhang et al., 2015*). Auxin administration at the first larval (L1) stage resulted in efficient depletion of *mNG::AID::cfi-1* expression, which was undetectable 2 days later (*Figure 5—figure supplement 2*). At the L3 stage (2 days upon continuous auxin treatment), we observed ectopic expression of *glr-4* in DA and DB neurons. These results suggest that CFI-1 is required post-embryonically to prevent DA and DB neurons from adopting mixed identity, underscoring the critical role of UNC-3 and Hox in maintaining *cfi-1* expression (*Figure 5G*).

## Minimal disruption of temporal modularity in UNC-3 function leads to locomotion defects

Animals carrying *unc-3* null alleles display severe locomotion defects (*Feng et al., 2020*), likely due to combined defects in the expression of UNC-3 targets from both 'initiation and maintenance' and 'maintenance-only' modules (*Figure 3C*). Genes from the 'initiation and maintenance' module include (among others) terminal identity genes coding for ACh biosynthesis components (*Figure 3A*). Hence, it is conceivable that loss of *unc-3* can lead to defects in ACh biosynthesis, likely contributing to locomotion defects. However, it is unclear whether, in cholinergic MNs, the maintained expression of any of the UNC-3 targets from the 'maintenance-only' module is critical for locomotion. To test this, we focused on *cfi-1/Arid3a* and used the CRISPR-engineered allele (*mNG::AID::cfi-1 $^{8\ COE\ motifs\ mut}$*) that selectively affects maintenance, but not initiation, of *cfi-1* expression in cholinergic MNs (*Figure 4E–F*). This allele minimally disrupts temporal modularity in UNC-3 function because UNC-3 can still control all its targets from both modules (*Figure 3C*), except one (*cfi-1*). As controls, we used animals carrying: (a) the endogenous *cfi-1* reporter allele (*mNG::AID::cfi-1*), (b) a putative null *cfi-1* allele (*ot786*) (*Kerk et al., 2017*), in which *cfi-1* activity is affected in MNs and other neuron types of the motor circuit (*Pereira et al., 2015*; *Shaham and Bargmann, 2002*), and (c) a deletion of the distal enhancer (*mNG::AID::cfi-1$^{\Delta enhancer(769\ bp)}$*), in which both initiation and maintenance of *cfi-1* are abrogated in nerve cord MNs (*Figure 6A*). We performed a high-resolution behavioral analysis of freely moving adult (day 2) animals of the above genotypes using automated multi-worm tracking technology (*Javer et al., 2018b*; *Yemini et al., 2013*). We found several features related to *C. elegans* locomotion (*e.g.,* body curvature, velocity) severely affected in *cfi-1 (ot786)* putative null animals (*Figure 6B–G*). Compared to *cfi-1* null mutants, animals carrying the *mNG::AID::cfi-1 $^{8\ COE\ motifs\ mut}$* allele (selective disruption of *cfi-1* maintenance) display milder, but statistically significant locomotion defects in the adult (*Figure 6B–G*). As expected, these defects were also present in animals carrying the *mNG::AID::cfi-1 $^{\Delta enhancer\ (769\ bp)}$* allele, in which both initiation and maintenance of *cfi-1* expression is affected. In summary, we specifically disrupted in cholinergic MNs the maintained expression of a single UNC-3 target (*cfi-1* from the 'maintenance-only' module) by using the *mNG::AID::cfi-1 $^{8\ COE\ motifs\ mut}$* allele and observed locomotion defects. This analysis suggests that minimal disruption of temporal modularity in UNC-3 function can affect animal behavior.

## Hox proteins and UNC-3 control *bnc-1/BNC* expression in cholinergic MNs

Our *cis*-regulatory analysis suggested that five TFs (*cfi-1/Arid3a, bnc-1/BNC1-2, mab-9/Tbx20, ceh-44/CUX1-2, nhr-40/nuclear hormone receptor*) require UNC-3 selectively for maintenance (*Figure 3B*). An in-depth analysis of *cfi-1/Arid3a* revealed that Hox proteins (LIN-39, MAB-5) and UNC-3 ensure the continuous expression of *cfi-1* in subsets of *unc-3*-positive MNs (*Figures 3–5*). We next asked whether a similar mechanism applies to the regulation of *bnc-1*/BNC, which is also expressed in a subset of *unc-3*-positive MNs (VA, VB) and prevents them from adopting a mixed identity (*Figure 5—figure supplement 3*; *Kerk et al., 2017*). Using an endogenous reporter allele (*bnc-1::mNG::AID*), we found that LIN-39 and MAB-5 are required for *bnc-1* expression at all stages examined (L2, L4, day 1 adult; *Figure 5—figure supplement 3*), suggesting a role for Hox in *bnc-1* initiation and maintenance. Next, we found that UNC-3 is absolutely required for *bnc-1* maintenance in the adult (day 1), albeit weaker effects were also observed at L2 and L4 (*Figure 5—figure supplement 3*). Similar to *cfi-1*, these findings strongly suggest that endogenous *bnc-1* expression depends on Hox and UNC-3.

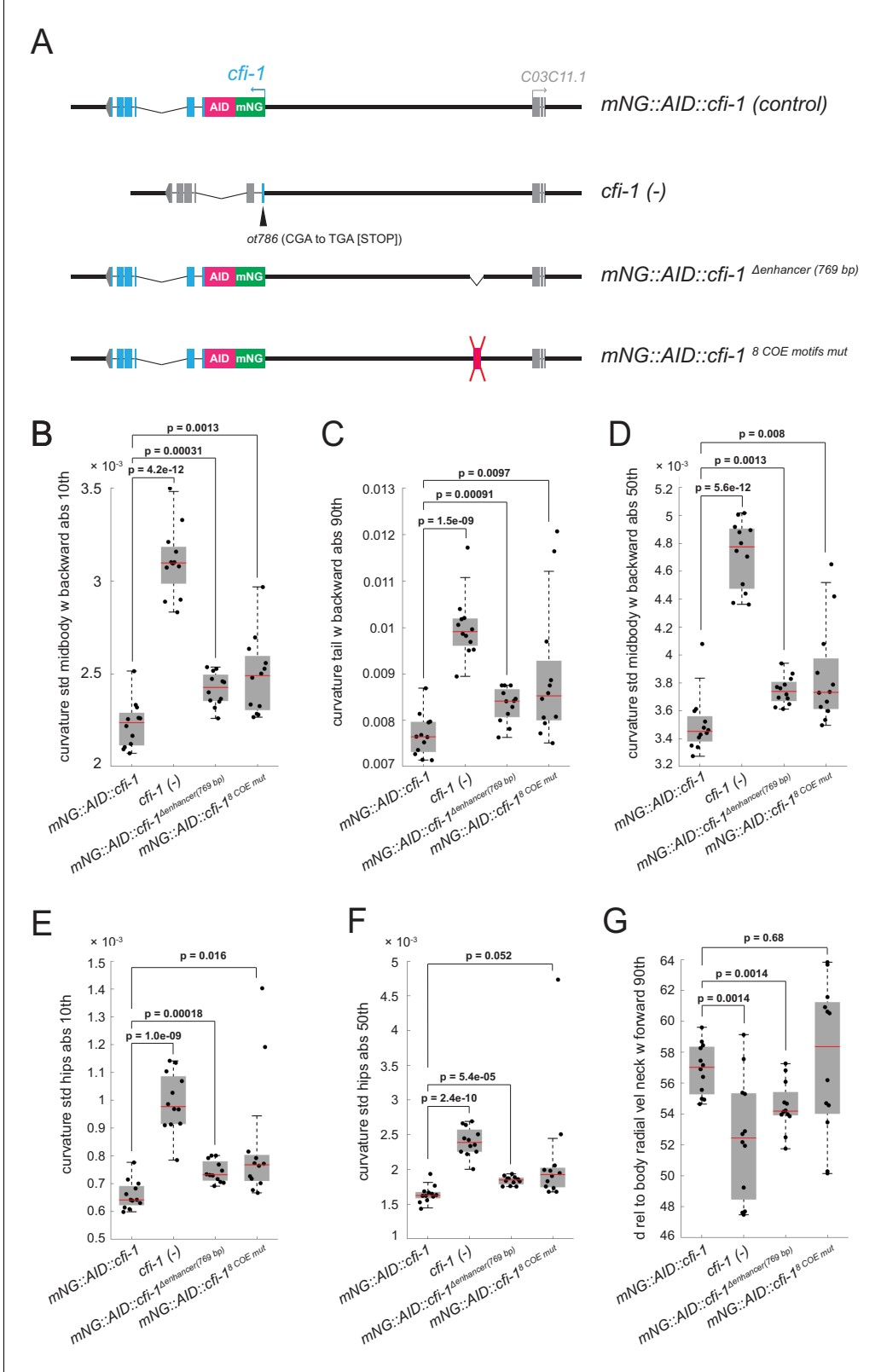

**Figure 6.** Minimal disruption of temporal modularity in UNC-3 function leads to locomotion defects. (**A**) Schematics illustrating four *cfi-1* alleles tested for behavioral analysis. (**B–G**) Examples of six locomotion features significantly disrupted in animals carrying a putative null (*ot786*) allele for *cfi-1*. Locomotion analysis was performed on day 2 adult worms. Animals lacking *cfi-1* expression (initiation and maintenance) specifically in MNs (*mNG::AID:: cfi-1* $^{\Delta enhancer\ (769\ bp)}$) and animals unable to maintain *cfi-1* expression in cholinergic MNs (*mNG::AID::cfi-1* $^{8\ COE\ motifs\ mut}$) display locomotion defects

*Figure 6 continued on next page*

*Figure 6 continued*

when compared to control *mNG::AID::cfi-1* animals. As expected, these defects are milder when compared to animals carrying the *cfi-1* (*ot786*) allele. Panels B-F show locomotion features related to body curvature, whereas panel G shows radial velocity of the neck. A detailed description of each locomotion feature is provided below. (B) curvature_std_mid-body_w_backward_abs_10th: 10th percentile of the absolute value of the standard deviation of the curvature of the mid-body, while worm is moving backwards. (C) curvature_tail_w_backward_abs_90th: 90th percentile of the absolute value of the curvature of the tail, while worm is moving backwards. (D) curvature_std_mid-body_w_backward_50th: 50th percentile of the standard deviation of the curvature of the mid-body, while worm is moving backwards. (E) curvature_std_hips_abs_10th: 10th percentile of the absolute value of the standard deviation of the curvature of the hips. (F) curvature_std_hips_abs_50th: 50th percentile of the absolute value of the standard deviation of the curvature of the hips. (G) d_rel_to_body_radial_vel_neck_w_forward_90th: 90th percentile of the derivative of radial velocity of the neck relative to the centroid of the mid-body points, while worm is moving forwards.

## Temporal modularity of UNC-30/PITX function in GABAergic MNs

Is temporal modularity observed in the function of other terminal selectors? To address this, we focused on UNC-30/PITX, the terminal selector of GABAergic MN (DD, VD) identity in the *C. elegans* nerve cord (*Figure 7A*; *Jin et al., 1994*). UNC-30/PITX is known to directly activate the expression of several terminal identity genes (e.g., *unc-25*/GAD [glutamic acid decarboxylase], *unc-47*/VGAT [vesicular GABA transporter])(*Eastman et al., 1999*), but a longitudinal analysis of target gene expression in *unc-30* null animals is lacking. Using reporter strains and methodologies similar to those used for UNC-3, we found that terminal identity gene (*unc-25*/GAD, *unc-47*/VGAT) expression is affected in GABAergic MNs of *unc-30* mutants at all stages examined (3-fold embryo, L2, L4, adult [day 1]) (*Figure 7B,D*), suggesting a requirement for initiation and maintenance. As mentioned earlier (*Figure 4C*), *cfi-1* is also expressed in GABAergic MNs (*Figure 7A*). Since UNC-3 is required for *cfi-1* maintenance in cholinergic MNs, we asked whether UNC-30/PITX plays a similar role to maintain *cfi-1* expression in GABAergic MNs. We found no effect at the embryonic stage (3-fold), but progressively stronger effects at larval (L2, L4) and adult (day 1) stages (*Figure 7C*), indicating that, in GABAergic MNs, UNC-30/PITX is selectively required for the maintenance, but not initiation of *cfi-1* expression. Taken together, our findings indicate that, like UNC-3 in cholinergic MNs (*Figure 3C*), the function of UNC-30 in GABAergic MNs is organized into two modules (module #1: initiation and maintenance; module #2: maintenance-only) (*Figure 7D–E*), suggesting temporal modularity may be a shared feature among terminal selector type-TFs.

To gain mechanistic insights, we analyzed available UNC-30 ChIP-Seq data at the L2 stage (*Yu et al., 2017*). UNC-30 binds to the *cis*-regulatory region of terminal identity genes (*unc-25*/GAD, *unc-47*/VAGT) (*Figure 7D*), confirming previous observations (*Eastman et al., 1999*). UNC-30 also binds to the same distal enhancer of *cfi-1* in GABAergic MNs, as UNC-3 does in cholinergic MNs (*Figure 7E*). However, the UNC-30 binding sites are distinct from the UNC-3 sites in this enhancer (*Figure 7C*). CRISPR-mediated deletion of this enhancer abolished *cfi-1* expression in both cholinergic and GABAergic MNs (*Figure 4A–B*). This finding suggests that maintenance of *cfi-1* expression in two different neuron types relies on the same enhancer receiving UNC-30/PITX input in GABAergic MNs and UNC-3/EBF input in cholinergic MNs. Interestingly, these results provide an example of 'enhancer pleiotropy' (*Sabarís et al., 2019*), in which the same *cis*-regulatory element is used to control gene expression in different neuron types.

## Discussion

Terminal selectors are continuously expressed TFs that determine the identity and function of individual neuron types (*Hobert, 2008*; *Hobert, 2016b*). However, the breadth of biological processes controlled by terminal selectors remains unclear. Moreover, whether terminal selectors control an identical suite of target genes across different life stages, or the suite of targets can change over time is largely unexplored. Filling such knowledge gaps can help us understand how cellular identity is established during development and maintained throughout life, a fundamental goal in the field of developmental biology. In this study, we focus on UNC-3/Ebf, the terminal selector of cholinergic MNs in the *C. elegans* nerve cord. Through ChIP-Seq, we identify in an unbiased manner the direct targets of UNC-3, uncovering the breadth of biological processes potentially controlled by this terminal selector. Unexpectedly, we find two groups of target genes with distinct temporal requirements for UNC-3 in cholinergic MNs. One group encodes different classes of proteins (e.g.,

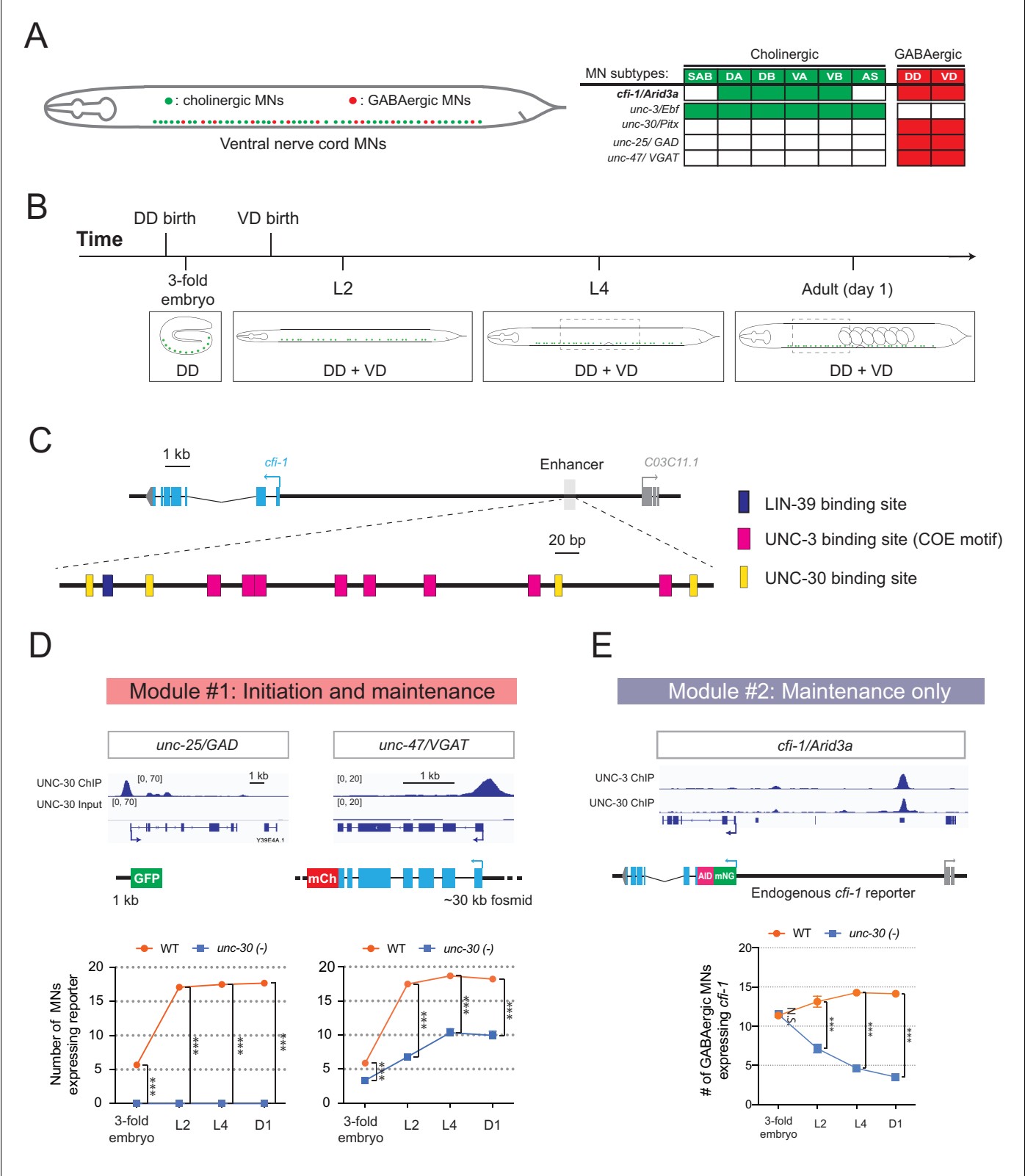

**Figure 7.** Temporal modularity in UNC-30/Pitx function in GABAergic MNs. (**A**) Schematic summarizing the expression of *cfi-1/Arid3a, unc-3/Ebf, unc-30/Pitx, unc-25/GAD,* and *unc-47/VGAT* in MN subtypes of the *C. elegans* ventral nerve cord. (**B**) Schematic showing time of birth and cell body position of GABAergic nerve cord MNs. DD neurons are born embryonically. VD neurons are born post-embryonically. (**C**) Bioinformatic analyses predict 4 UNC-30 binding sites (yellow boxes) in the *cfi-1* enhancer. The location of UNC-3 and LIN-39 binding sites are also shown. (**D**) Top: snapshots

*Figure 7 continued on next page*

*Figure 7 continued*

of UNC-30 ChIP-Seq and input (negative control) signals at the *cis*-regulatory regions of 2 GABAergic terminal identity genes (*unc-25/GAD, unc-47/VGAT*). Bottom: quantification of the expression of transgenic reporters in WT and *unc-30 (e191)* animals at four different developmental stages – 3-fold embryo, L2, L4, and day 1 adults (N = 15). UNC-30 is required for both initiation and maintenance of *unc-25/GAD* and *unc-47/VGAT*. ***p<0.001. (**E**) Top: a snapshot of UNC-3 ChIP-Seq and UNC-30 ChIP-Seq signals at the *cfi-1* locus. Bottom: quantification of the number of MNs expressing the endogenous reporter *mNG::AID::cfi-1* in WT and *unc-30 (e191)* animals. All *cfi-1*-expressing MNs in the ventral cord (cholinergic and GABAergic MNs) were counted in 3-fold embryos due to a lack of a specific marker that labels GABAergic MNs in embryos. Expression of *cfi-1* specifically in GABA neurons was quantified at L2, L4, and day 1 adult stages (N ≥ 12). At those stages, cholinergic MNs were identified based on a fluorescent marker (*cho-1::mChOpti*), which are ruled out during scoring. GABAergic MNs were scored positive for *cfi-1* expression when the *mNG::AID::cfi-1* (green) expression co-localized with *ttr-39::mCherry* (red). N.S.: not significant, ***p<0.001.

receptors, transporters) that require UNC-3 activation at all life stages examined. However, a second group exclusively encodes MN-specific TFs that selectively require UNC-3 for maintenance of their expression at late larval and adult stages. Hence, the suite of UNC-3 targets in cholinergic MNs is partially modified across different life stages, a phenomenon we term 'temporal modularity' in terminal selector function. Minimal disruption of this modularity by selectively removing the ability of UNC-3 to maintain expression of a single target gene (the TF *cfi-1/Arid3a* from the second group) led to locomotion defects, highlighting the necessity of temporal modularity for animal behavior.

## Temporal modularity in terminal selector function may represent a general principle for neuronal subtype diversity

Why is there a need for temporal modularity in the function of continuously expressed TFs, such as terminal selectors? The case of UNC-3 suggests that temporal modularity is necessary for generating and maintaining neuronal subtype diversity. UNC-3 is continuously expressed in six MN subtypes (SAB, DA, DB, VA, VB, AS) of the *C. elegans* ventral nerve cord (*Figure 4C*; *Pereira et al., 2015*). However, most UNC-3 targets from either module (modules # 1 and #2, *Figure 3C*) are expressed in some, but not all, of these subtypes. By ensuring the continuous expression of genes from module #1 ('initiation and maintenance' module, *Figure 3C*), such as terminal identity genes [(*Kratsios et al., 2011*) and this study], UNC-3 consolidates the unique identity of each MN subtype. However, target genes from module #2 ('maintenance-only' module, *Figure 3C*) escape initiation by UNC-3, but do require UNC-3 for maintenance. Interestingly, all module #2 targets code for TFs (*cfi-1/Arid3a, bnc-1/BNC1-2, mab-9/Tbx20, ceh-44/Cux1, nhr-40*), specifically expressed in subsets of the six subtypes (SAB, DA, DB, VA, VB, AS) (*Kerk et al., 2017* and this study). Three of these factors (*cfi-1/Arid3a, bnc-1/Bnc1, mab-9/Tbx20*) are thought to act as transcriptional repressors to prevent the adoption of mixed MN identity (*Kerk et al., 2017*). For example, loss of *cfi-1* during early development results in 'mixed' identity of DA and DB subtypes, as these cells, in addition to their normal molecular signature, also acquire expression of genes (e.g., *glr-4/GluR*) normally expressed in other MNs (*Figure 5—figure supplement 2*). Through protein depletion experiments at post-embryonic stages, we show here that CFI-1 is continuously required to prevent DA and DB neurons from acquiring a mixed identity. Hence, UNC-3 indirectly maintains the unique identity of individual MN subtypes (DA, DB) by selectively maintaining expression of the TF *cfi-1* (from module #2) during late larval and adult stages. In addition, UNC-3 acts directly to control DA and DB identity by ensuring the continuous expression of *unc-129/TGFb*, a DA- and DB-specific terminal identity gene (from module #1) (*Figure 3A*, *Figure 5—figure supplement 2*). Hence, temporal modularity in UNC-3 function can be envisioned as a 'double safety' mechanism for the generation and maintenance of MN diversity.

The mechanism of temporal modularity may represent a general paradigm of how terminal selectors establish and maintain subtype identity within a class of functionally related neurons. Indeed, the same terminal selector is often continuously expressed in multiple subtypes of a given neuronal class (*Hobert and Kratsios, 2019*). In mice, for example, the terminal selectors Nurr1 and Pet-1 are expressed in several subtypes of dopaminergic and serotonergic neurons, respectively (*Kadkhodaei et al., 2009*; *Okaty et al., 2015*). However, future studies are needed to determine whether temporal modularity in the function of these terminal selectors is necessary for establishing and maintaining neuronal subtype identity.

## Temporal modularity offers key insights into how terminal selectors control neuronal identity over time

The prevailing hypothesis of how terminal selectors establish and maintain neuronal identity is that they bind constantly, from development through adulthood, on the *cis*-regulatory region of terminal identity genes, and thereby continuously activate their expression (*Deneris and Hobert, 2014*; *Hobert, 2008*; *Hobert and Kratsios, 2019*). For most terminal selectors, however, biochemical evidence for binding and longitudinal analysis of terminal selector mutant animals are currently lacking. Our analysis on module #1 genes (e.g., terminal identity genes) supports the aforementioned hypothesis; ChIP-Seq for UNC-3 demonstrated binding on these genes (*Figure 3A,C*). Moreover, our longitudinal genetic analysis combined with the UNC-3 protein depletion experiments in adult MNs demonstrate that UNC-3 is continuously required to maintain terminal identity gene expression.

Interestingly, the analysis on module #2 genes provides new mechanistic insights into how terminal selectors control neuronal identity over time. Instead of constantly activating the same set of genes, as predicted by the above hypothesis, terminal selectors can also partially modify the suite of their target genes at different life stages (temporal modularity), suggesting their function can be more dynamic than previously thought. This is based on the finding that UNC-3 is selectively required for maintenance, not initiation, of five conserved TF-encoding genes from module #2 (*cfi-1/ Arid3a, bnc-1/BNC1-2, mab-9/Tbx20, ceh-44/Cux1, nhr-40*) (*Figure 3B–C*). Hence, some targets (module #1) require constant UNC-3 input from development through adulthood, whereas others (module #2) require UNC-3 only for maintenance.

By honing in on one TF (*cfi-1/Arid3a)* downstream of UNC-3, we identify a mechanism that potentially enables initiation and maintenance of module #2 genes (*Figure 5G*). That is, the Hox proteins LIN-39 and MAB-5 control initiation of *cfi-1/Arid3a* in cholinergic MNs independently of UNC-3, but *cfi-1* maintenance in these same neurons depends on both Hox and UNC-3 (*Figure 5G*). Mechanistically, we propose that Hox-dependent initiation and Hox/UNC-3-dependent maintenance of *cfi-1* are 'funneled' through the same distal enhancer, which bears both Hox and UNC-3 binding sites. This suggests embryonic initiation and post-embryonic maintenance of expression of a particular gene, in a specific cell type, can be achieved by distinct TF combinations acting upon the same *cis*-regulatory region (enhancer) (*Figure 5G*). This somewhat surprising mechanism differs from previous fly and mouse studies reporting distinct and physically separated *cis*-regulatory regions necessary for either initiation or maintenance of cell type-specific gene expression (*Ellmeier et al., 2002*; *Johnson et al., 2011*; *Manzanares et al., 2001*; *Pfeffer et al., 2002*; *Rhee et al., 2016*).

In summary, our findings critically extend the mechanisms underlying UNC-3 function. Previous work demonstrated that, in cholinergic MNs, UNC-3 not only activates expression of terminal identity genes (*Kratsios et al., 2011*), but also prevents alternative neuronal identities (*Feng et al., 2020*). Here, we report that the suite of UNC-3 targets in these neurons can be partially modified at different life stages, offering key insights into how terminal selectors control neuronal identity over time.

## Hox proteins collaborate with stage-specific TFs to establish and maintain MN terminal identity

During early neuronal development, Hox proteins are essential for cell survival, neuronal diversity, and circuit assembly (*Baek et al., 2013*; *Catela et al., 2016*; *Estacio-Gómez and Díaz-Benjumea, 2014*; *Karlsson et al., 2010*; *Miguel-Aliaga and Thor, 2004*; *Moris-Sanz et al., 2015*; *Philippidou and Dasen, 2013*). However, their post-embryonic neuronal functions remain elusive. Moreover, Hox proteins are often continuously expressed in multiple cell types of a given body region (*Baek et al., 2013*; *Hutlet et al., 2016*; *Takahashi et al., 2004*), raising the question of how do they achieve cell type-specificity in their function. Our findings on mid-body Hox proteins LIN-39 and MAB-5 begin to address this question.

LIN-39 and MAB-5 are continuously expressed in multiple cell types located at the *C. elegans* mid-body region (*Clark et al., 1993*; *Cowing and Kenyon, 1992*; *Feng et al., 2020*; *Maloof and Kenyon, 1998*). We find that LIN-39 and MAB-5 exert a cell type-specific function; they are required to initiate (in embryo) and maintain (post-embryonically) *cfi-1* expression in specific subsets of cholinergic MNs (DA, DB, VA, VB; *Figure 4C*). Such specificity likely arises through collaboration with

distinct TFs responsible for either initiation or maintenance of *cfi-1* (*Figure 5G*). Supporting this scenario, Hox (LIN-39, MAB-5) and UNC-3 are co-expressed in DA, DB, VA, and VB neurons (*Feng et al., 2020*). Moreover, UNC-3 is selectively required for *cfi-1* maintenance (not initiation) in these neurons (*Figure 5G*). We surmise that other, yet-to-be-identified factors collaborate with Hox proteins at early stages to initiate *cfi-1* expression specifically in DA, DB, VA, and VB neurons. Such initiation-specific factors likely act through the same distal enhancer because its deletion completely abolished *cfi-1* expression in these neurons at early (and late) stages (*Figures 4D* and *5G*). In summary, we propose that Hox proteins collaborate with distinct TFs over time, that is, initiation-specific factors and the terminal selector UNC-3, to ensure continuous expression of *cfi-1/Arid3a* in specific subtypes of cholinergic MNs. This mechanism may extend to the regulation of terminal identity genes, as we previously showed that LIN-39 and UNC-3 are required to maintain expression of *acr-2/* human CHRNA1 (acetylcholine receptor subunit) and *unc-77/* human NALCN (sodium channel) in cholinergic MNs (*Feng et al., 2020*). Altogether, these findings offer mechanistic insights into the recently proposed hypothesis that Hox proteins in *C. elegans* collaborate with terminal selectors to establish and maintain neuronal terminal identity (*Kratsios et al., 2017*; *Zheng et al., 2015*).

## Limitations of this study

By conducting a longitudinal analysis for 14 UNC-3 target genes, we identified two groups with distinct temporal requirements (modules #1 and #2 in *Figure 3C*). It is likely though that UNC-3 temporally controls other targets through additional modules. For example, animals lacking *unc-3* display severe axon guidance defects in cholinergic MNs (*Prasad et al., 1998*), but the underlying mechanisms remain unknown. Since axon guidance molecules are often expressed in a transient fashion during early neuronal development, we speculate that UNC-3 may transiently activate expression of such molecules, a possibility that would add another temporal module in UNC-3 function. Another likely scenario is an 'initiation-only' module where UNC-3 is responsible only for onset, but not maintenance, of a yet-to-be-identified set of targets.

In addition, the breadth of biological processes controlled by terminal selectors remains largely unknown. Our ChIP-Seq analysis potentially implicates UNC-3 in a range of biological processes. First, this dataset significantly extends previous reports on the role of UNC-3 in neuronal terminal identity (*Kim et al., 2005*; *Kratsios et al., 2011*) by identifying hundreds of terminal identity genes (42.18% of total ChIP-Seq hits) as putative UNC-3 targets. Second, the ChIP-Seq dataset suggests new roles for UNC-3 in neuronal metabolic pathways (24.14% of UNC-3 target genes code for enzymes) and gene regulatory networks (24.07% of UNC-3 targets are TFs and nucleic acid-binding proteins). However, future RNA-Sequencing studies in *unc-3*-depleted MNs are necessary to correlate gene expression changes with UNC-3 ChIP-Seq data, and thereby identify *bona fide* targets and biological processes under direct UNC-3 control. Such analysis may also uncover any indirect effects on gene expression in *unc-3*-depleted MNs, perhaps arising due to a partial cell fate transformation previously reported for these neurons (*Feng et al., 2020*).

## Temporal modularity may be a shared feature among continuously expressed TFs

This study suggests that the suite of targets of two *C. elegans* terminal selectors (UNC-3/Ebf and UNC-30/Pitx) can be modified over time, providing evidence for temporal modularity in their function. Given that terminal selectors, as well as other neuron type-specific TFs with continuous expression, have been described in both invertebrate and vertebrate species (*Deneris and Hobert, 2014*; *Hobert and Kratsios, 2019*; *Mayer et al., 2018*; *Mi et al., 2018*), it will be interesting to determine the potential generality of the temporal mechanism described here. Supporting this possibility, the terminal selector of serotonergic neurons in mice (Pet-1) activates expression of serotonin biosynthesis proteins during development, but appears to switch transcriptional targets at later life stages (*Wyler et al., 2016*). Outside the nervous system, cell type-specific TFs with continuous expression have been described in worms, flies and mice (*Pikkarainen et al., 2004*; *Soler et al., 2012*; *Wiesenfahrt et al., 2016*; *Zhou et al., 2017*). Future studies will determine whether the principle of temporal modularity is widely employed for the control of cell type identity.

# Materials and methods

**Key resources table**

| Reagent type (species) or resource | Designation | Source or reference | Identifiers | Additional information |
|---|---|---|---|---|
| Gene (*Caenorhabditis elegans*) | *unc-3* | Wormbase | WBGene00006743 | |
| Gene (*C. elegans*) | *cfi-1* | Wormbase | WBGene00000476 | |
| Gene (*C. elegans*) | *unc-30* | Wormbase | WBGene00006766 | |
| Gene (*C. elegans*) | *lin-39* | Wormbase | WBGene00003024 | |
| Gene (*C. elegans*) | mab-5 | Wormbase | WBGene00003102 | |
| Strain, strain background (*C. elegans*) | *unc-3 (n3435)* | Bob Horvitz (MIT, Cambridge MA) | MT10785 | Null Allele: deletion |
| Strain, strain background (*C. elegans*) | *unc-3 (e151)* | Caenorhabditis Genetics Center | CB151 | Allele: substitution |
| Strain, strain background (*C. elegans*) | *lin-39 (n1760) mab-5 (e1239)/ht2 III; lgIs58 [[gcy-32::gfp]] V* | Caenorhabditis Genetics Center | LE4023 | |
| Strain, strain background (*C. elegans*) | *lin-39 (n1760)/dpy-17 (e164) unc-32 (e189) III* | Caenorhabditis Genetics Center | MT4009 | Null Allele: substitution |
| Strain, strain background (*C. elegans*) | *unc-30 (e191)* | Caenorhabditis Genetics Center | CB845 | Allele: substitution |
| Strain, strain background (*C. elegans*) | *cfi-1 (ot786)* | This paper. Kratsios lab (University of Chicago, IL, USA). | KRA464 | Null Allele: substitution - R to STOP (39); Strain was 3x backcrossed |
| Strain, strain background (*C. elegans*) | *unc-3 (ot839 [unc-3::gfp]) X* | Oliver Hobert (Columbia University, New York NY) | OH13990 | CRISPR-generated allele |
| Strain, strain background (*C. elegans*) | *cfi-1 (kas16 [mNG::AID::cfi-1]) I* | This paper. Kratsios lab (University of Chicago, IL, USA). | KRA345 | CRISPR-generated allele; See Materials and methods - Targeted genome editing |
| Strain, strain background (*C. elegans*) | *bnc-1 (ot845 [bnc-1::mNGAID]) V* | **Kerk et al., 2017** | OH14070 | CRISPR-generated allele |
| Strain, strain background (*C. elegans*) | *lin-39 (kas9 [lin-39::mNG::AID]) III* | **Feng et al., 2020** Jan 3. doi:10.7554/eLife.50065 | KRA467 | CRISPR-generated allele |
| Strain, strain background (*C. elegans*) | *unc-3 (ot837 [unc-3::mNG::AID]) X; ieSi57 [Peft-3::TIR1:: mRuby::unc-54 3' UTR, cb-unc-119(+)] II* | This paper. Kratsios lab (University of Chicago, IL, USA). | KRA376 | CRISPR-generated allele |
| Strain, strain background (*C. elegans*) | *cfi-1 (syb1812 [A4e_enhancer deletion −13,264–12,495 from cfi-1 ATG in kas16] kas16[cfi-1::mNG::AID]) I* | This paper. Kratsios lab (University of Chicago, IL, USA). | PHX1812 | See Materials and methods - Targeted genome editing, and ***Figure 4A*** legend |
| Strain, strain background (*C. elegans*) | *cfi-1 (syb1856 [8 COE motifs mut in A4e] kas16 [cfi-1::mNG::AID]) I* | This paper. Kratsios lab (University of Chicago, IL, USA). | PHX1856 | See Materials and methods - Targeted genome editing, and ***Figure 4E*** legend |

*Continued on next page*

*Continued*

| Reagent type (species) or resource | Designation | Source or reference | Identifiers | Additional information |
|---|---|---|---|---|
| Strain, strain background (*C. elegans*) | *otIs544 [cho-1(fosmid):: SL2::mCherry::H2B + pha-1(+)]* | Oliver Hobert (Columbia University, New York NY) | OH13646 | |
| Strain, strain background (*C. elegans*) | *juIs14 [acr-2p::GFP + lin-15(+)] IV.* | Caenorhabditis Genetics Center | CZ631 | |
| Strain, strain background (*C. elegans*) | *otIs426 [Punc-17_1 kb:: YFP, Pmyo-2::GFP]* | Oliver Hobert (Columbia University, New York NY) | OH11454 | |
| Strain, strain background (*C. elegans*) | *evIs82b [unc-129:: GFP + dpy-20(+)] IV.* | Oliver Hobert (Columbia University, New York NY) | OH1894 | |
| Strain, strain background (*C. elegans*) | *otIs476 [glr-4 prom:: TagRFP + pha-1(+)]* | **Kerk et al., 2017** | OH12052 | |
| Strain, strain background (*C. elegans*) | *otIs477 [glr-4 prom:: TagRFP + pha-1(+)]* | Oliver Hobert (Columbia University, New York NY) | OH12053 | |
| Strain, strain background (*C. elegans*) | *vsIs48 [unc-17::GFP]* | Caenorhabditis Genetics Center | LX929 | |
| Genetic reagent (*C. elegans*) | *Pcfi-1_8.17 kb::GFP* | **Shaham and Bargmann, 2002** | nsEx37 | See *Figure 4A* legend |
| Genetic reagent (*C. elegans*) | *cfi-1_−11,329 bp −13,824 bp::RFP* | This paper. Kratsios lab (University of Chicago, IL, USA). | kasEx12 kasEx182 kasEx183 | See Materials and methods, and *Figure 4A* legend |
| Genetic reagent (*C. elegans*) | *cfi-1_2.77 kb(993 bp to 3,764 bp)::rfp* | This paper. Kratsios lab (University of Chicago, IL, USA). | kasEx184 kasEx185 kasEx186 | See Materials and methods, and *Figure 4A* legend |
| Genetic reagent (*C. elegans*) | *cfi-1_1.72 kb (547 bp to −1,173 bp)::rfp* | This paper. Kratsios lab (University of Chicago, IL, USA). | kasEx187 kasEx188 kasEx189 | See Materials and methods, and *Figure 4A* legend |
| Genetic reagent (*C. elegans*) | *cfi-1_1.71 kb (−1,164 bp to −2,875 bp): :rfp* | This paper. Kratsios lab (University of Chicago, IL, USA). | kasEx190 kasEx191 kasEx192 | See Materials and methods, and *Figure 4A* legend |
| Genetic reagent (*C. elegans*) | *cfi-1_3.27 kb(−2,865 bp to −6,141 bp)::rfp* | This paper. Kratsios lab (University of Chicago, IL, USA). | kasEx193 kasEx194 kasEx195 | See Materials and methods, and *Figure 4A* legend |
| Genetic reagent (*C. elegans*) | *cfi-1_3.18 kb(−8,162 bp to −11,346 bp): :rfp* | This paper. Kratsios lab (University of Chicago, IL, USA). | kasEx196 kasEx197 | See Materials and methods, and *Figure 4A* legend |
| Genetic reagent (*C. elegans*) | *cfi-1_552 bp(−11,329 bp to −11,881 bp): :rfp* | This paper. Kratsios lab (University of Chicago, IL, USA). | kasEx198 kasEx199 kasEx200 | See Materials and methods, and *Figure 4A* legend |
| Genetic reagent (*C. elegans*) | *cfi-1_383 bp(−11,851 bp to −12,234 bp): :rfp* | This paper. Kratsios lab (University of Chicago, IL, USA). | kasEx201 kasEx202 kasEx203 | See Materials and methods, and *Figure 4A* legend |
| Genetic reagent (*C. elegans*) | *cfi-1_499 bp(−12,223 bp to −12,722 bp)::rfp* | This paper. Kratsios lab (University of Chicago, IL, USA). | kasEx204 kasEx205 | See Materials and methods, and *Figure 4A* legend |

*Continued on next page*

*Continued*

| Reagent type (species) or resource | Designation | Source or reference | Identifiers | Additional information |
|---|---|---|---|---|
| Genetic reagent (*C. elegans*) | *cfi-1_579 bp(−12,705 bp to −13,284 bp)::rfp* | This paper. Kratsios lab (University of Chicago, IL, USA). | kasEx206 kasEx207 kasEx208 | See Materials and methods, and *Figure 4A* legend |
| Genetic reagent (*C. elegans*) | cfi-1_561 bp(−13,263 bp to −13,824 bp)::rfp | This paper. Kratsios lab (University of Chicago, IL, USA). | kasEx209 kasEx210 | See Materials and methods, and *Figure 4A* legend |
| Genetic reagent (*C. elegans*) | *nhr-1 (peak 1,−7455 to −5853)::RFP:: unc-54 3'UTR* | This paper. Kratsios lab (University of Chicago, IL, USA). | kasEx211 | See Materials and methods - Generation of transgenic animals carrying transcriptional fusion reporters |
| Genetic reagent (*C. elegans*) | *nhr-1 (peak 2,−893 to +1535)::RFP:: unc-54 3'UTR* | This paper. Kratsios lab (University of Chicago, IL, USA). | kasEx212 kasEx213 | See Materials and methods - Generation of transgenic animals carrying transcriptional fusion reporters |
| Genetic reagent (*C. elegans*) | *nhr-40 (peak 1, +938 to +1846)::RFP:: unc-54 3'UTR* | This paper. Kratsios lab (University of Chicago, IL, USA). | kasEx214 kasEx215 | See Materials and methods - Generation of transgenic animals carrying transcriptional fusion reporters |
| Genetic reagent (*C. elegans*) | *nhr-40 (peak 2, +4360 to +5522)::RFP:: unc-54 3'UTR* | This paper. Kratsios lab (University of Chicago, IL, USA). | kasEx216 kasEx217 | See Materials and methods - Generation of transgenic animals carrying transcriptional fusion reporters |
| Genetic reagent (*C. elegans*) | *zfh-2::rfp (12,048 bp to 13,549 bp)* | This paper. Kratsios lab (University of Chicago, IL, USA). | kasEx218 kasEx219 | See Materials and methods - Generation of transgenic animals carrying transcriptional fusion reporters |
| Genetic reagent (*C. elegans*) | *nhr-49 (−803 to +58):: RFP::unc-54 3'UTR* | This paper. Kratsios lab (University of Chicago, IL, USA). | kasEx220 kasEx221 | See Materials and methods - Generation of transgenic animals carrying transcriptional fusion reporters |
| Genetic reagent (*C. elegans*) | *nhr-19 (+1250 to +2302)::RFP::unc-54 3'UTR* | This paper. Kratsios lab (University of Chicago, IL, USA). | kasEx222 kasEx223 | See Materials and methods - Generation of transgenic animals carrying transcriptional fusion reporters |
| Genetic reagent (*C. elegans*) | *ccch-3 (Peak 1, 0 to −387)::RFP:: unc-54 3'UTR* | This paper. Kratsios lab (University of Chicago, IL, USA). | kasEx224 kasEx225 | See Materials and methods - Generation of transgenic animals carrying transcriptional fusion reporters |
| Genetic reagent (*C. elegans*) | *ztf-17 (Peak 1, 0 to −1034)::RFP:: unc-54 3'UTR* | This paper. Kratsios lab (University of Chicago, IL, USA). | kasEx226 kasEx227 | See Materials and methods - Generation of transgenic animals carrying transcriptional fusion reporters |
| Genetic reagent (*C. elegans*) | *ztf-26 (Peak 1,−1341 to +753)::RFP:: unc-54 3'UTR* | This paper. Kratsios lab (University of Chicago, IL, USA). | kasEx228 kasEx229 | See Materials and methods - Generation of transgenic animals carrying transcriptional fusion reporters |
| Genetic reagent (*C. elegans*) | *ztf-26 (Peak 2, +909 to +3394)::RFP:: unc-54 3'UTR* | This paper. Kratsios lab (University of Chicago, IL, USA). | kasEx230 kasEx231 | See Materials and methods - Generation of transgenic animals carrying transcriptional fusion reporters |
| Genetic reagent (*C. elegans*) | *mab-9 (−5569 to −3768)::RFP::unc-54 3'UTR* | This paper. Kratsios lab (University of Chicago, IL, USA). | kasEx232 kasEx233 | See Materials and methods - Generation of transgenic animals carrying transcriptional fusion reporters |
| Genetic reagent (*C. elegans*) | *ceh-44 (1,605 bp to 3,111 bp): :rfp* | This paper. Kratsios lab (University of Chicago, IL, USA). | kasEx234 kasEx235 | See Materials and methods - Generation of transgenic animals carrying transcriptional fusion reporters |

*Continued on next page*

*Continued*

| Reagent type (species) or resource | Designation | Source or reference | Identifiers | Additional information |
|---|---|---|---|---|
| Antibody | anti-GFP (Rabbit polyclonal) | Abcam | Ab290 | Dilution: 1:1000; RRID:AB_303395 |
| Commercial assay or kit | Dynabeads Protein G | Invitrogen | 1004D | |
| Commercial assay or kit | Gibson Assembly Cloning Kit | NEB | #5510S | |
| Commercial assay or kit | QIAquick PCR Purification Kit | QIAGEN | #28104 | |
| Commercial assay or kit | Ampure XP beads | Beckman Coulter Life Sciences | A63881 | |
| Chemical compound, drug | cOmplete ULTRA Protease Inhibitor Cocktail | Roche | #05892970001 | |
| Chemical compound, drug | Auxin (indole-3-acetic acid) | Alfa Aesar | #10196875 | |
| Software, algorithm | ZEN | ZEISS | Version 2.3.69.1000, Blue edition | RRID:SCR_013672 |
| Software, algorithm | Image J | Image J | Version 1.52i | RRID:SCR_003070 |
| Software, algorithm | Adobe Photoshop CS6 | Adobe | Version 13.0 × 64 | |
| Software, algorithm | Adobe Illustrator CS6 | Adobe | Version 16.0.0 × 64 | |

## *C. elegans* strain culture

Worms were grown at 20C or 25°C on nematode growth media (NGM) plates supplied with *E. coli* OP50 as food source (*Brenner, 1974*). A list of all *C. elegans* strains used is provided in Key Resources Table.

## Generation of transgenic animals carrying transcriptional fusion reporters

Reporter gene fusions for *cis*-regulatory analyses and validation of newly identified UNC-3 target genes were made with PCR fusion (*Hobert, 2002*). Genomic regions were amplified and fused to the coding sequence of *tagrfp* followed by the *unc-54* 3′ UTR. To mutate the LIN-39 binding motif, the reporter fusion was first introduced into the pCR-XL-TOPO vector using the TOPO XL PCR cloning kit (Invitrogen). Then, mutagenesis PCR was performed, and single clones containing plasmids that carry the desired mutation were isolated. PCR fusion DNA fragments were injected into young adult *pha-1(e2123)* hermaphrodites at 50 ng/µl together with *pha-1* (pBX plasmid) as co-injection marker (50 ng/µl).

## Chromatin immunoprecipitation (ChIP)

ChIP assay was performed as previously described (*Yu et al., 2017*; *Zhong et al., 2010*) with the following modifications. Synchronized L1 *unc-3 (ot839 [unc-3::gfp])* worms and N2 worms were cultured on 10 cm plates seeded with OP50 at 20°C overnight. Late L2 worms were cross-linked and resuspended in FA buffer supplemented with protease inhibitors (150 mM NaCl, 10 µl 0.1 M PMSF, 100 µl 10% SDS, 500 µl 20% N-Lauroyl sarsosine sodium, 2 tablets of cOmplete ULTRA Protease Inhibitor Cocktail [Roche Cat.# 05892970001] in 10 ml FA buffer). For each IP experiment, 200 µl worm pellet was collected. The sample was then sonicated using a Covaris S220 at the following settings: 200 W Peak Incident Power, 20% Duty Factor, 200 Cycles per Burst for 1 min. Samples were transferred to centrifuge tubes and spun at the highest speed for 15 min. The supernatant was transferred to a new tube, and 5% of the material was saved as input and stored at −20°C. The remainder was incubated with 2 µl GFP antibody (Abcam Cat.# ab290) at 4°C overnight. Wild-type (N2) worms do not carry the GFP tag and serve as negative control. The *unc-3 (ot839 [unc-3::gfp])* CRIPSR generated allele was used in order to immunoprecipitate the endogenous UNC-3 protein. On the next day, 20 µl Dynabeads Protein G (1004D) was added to the immunocomplex, which was then incubated for 2

hr at 4°C. The beads then were washed at 4°C twice with 150 mM NaCl FA buffer (5 min each), and once with 1M NaCl FA buffer (5 min). The beads were transferred to a new centrifuge tube and washed twice with 500 mM NaCl FA buffer (10 min each), once with TEL buffer (0.25 M LiCl, 1% NP-40, 1% sodium deoxycholate, 1 mM EDTA, 10 mM Tris-HCl, pH 8.0) for 10 min, and twice with TE buffer (5 min each). The immunocomplex was then eluted in 200 μl elution buffer (1% SDS in TE with 250 mM NaCl) by incubating at 65°C for 20 min. The saved input samples were thawed and treated with the ChIP samples as follows. One (1) μl of 20 mg/ml proteinase K was added to each sample and the samples were incubated at 55°C for 2 hr then 65°C overnight (12–20 hr) to reverse cross-link. The immonuprecipitated DNA was purified with Ampure XP beads (A63881) according to manufacturer's instructions.

### ChIP-sequencing data analysis

Unique reads were mapped to the *C. elegans* genome (ce10) with bowtie2 (*Langmead and Salzberg, 2012*). Peak calling was then performed with MACS2 (minimum q-value cutoff for peak detection: 0.005). For visualization purposes, the sequencing depth was normalized to 1x genome coverage using bamCoverage provided by deepTools (*Ramírez et al., 2016*), and peak signals were shown in IGV. Heatmap of peak coverage in regard to UNC-3 enrichment center was generated with NGSplot (*Shen et al., 2014*). The average profile of peaks binding to TSS region was generated with ChIPseeker (*Yu et al., 2015*). To study the distribution of peaks genome-wide, the peaks were annotated using annotatePeaks.pl provided by Homer (*Heinz et al., 2010*), and each peak was assigned to a gene with the nearest TSS. For de novo motif discovery, sequences containing 100 bp around the centers of each peak (from −50 bp to +50 bp) were extracted and supplied to findMotifsGenome.pl provided by Homer.

### Protein class ontology analysis using PANTHER

Protein class ontology analysis was performed on 1,478 UNC-3-bound genes out of the 3502 protein-coding genes. The number of genes is significantly lower than the number of peaks because PANTHER analysis only considers genes with known protein class terms.

### Targeted genome editing

The *cfi-1* endogenous reporter strain *kas16 [mNG::AID::cfi-1]* was generated by employing CRISPR/Cas9 genome editing, inserting the *mNG::3xFLAG::AID* cassette immediately after the ATG of *cfi-1*. The *cfi-1* enhancer knock-out allele *mNG::AID::cfi-1$^{\Delta enhancer\ (769\ bp)}$* was generated by using two guide RNAs flanking the *cfi-1* enhancer to guide excision of the genomic region, which was then followed by homology dependent repair (HDR) to create a 769 bp deletion (−11,329 bp to −12,097 bp). The UNC-3 binding motif mutation allele *mNG::AID::cfi-1$^{8\ COE\ motifs\ mut}$* was generated by creating nucleotide substitutions in the repair template, which carries homology arms complementary to the *cfi-1* enhancer region and is then introduced into the genome through HDR.

### Microscopy

Imaging slides were prepared by anesthetizing worms with sodium azide (NaN$_3$, 100 mM) and mounting them on a 4% agarose pad on glass slides. Images were taken with an automated fluorescence microscope (Zeiss, Axio Imager Z2). Images containing several Z-stacks (0.50 μm intervals between stacks) were taken with Zeiss Axiocam 503 mono using the ZEN software (Version 2.3.69.1000, Blue edition). Representative images are shown following max-projection of 2–5 μm Z-stacks using the maximum intensity projection type. Image reconstruction was performed with Image J (*Schindelin et al., 2012*).

### MN subtype identification

The identification of specific MN subtypes expressing a given UNC-3 target gene was assessed based on the following: (a) co-localization with fluorescent reporters that label specific MN subtypes; (b) Invariant position of neuronal cell bodies along the ventral nerve cord; (c) Birth order of specific MN subtypes (e.g. during embryogenesis or post-embryogenesis); (d) Total cell numbers in each MN subtype.

### Bioinformatic prediction of binding motifs

Information of the LIN-39 binding motif is curated in the Catalog of Inferred Sequence Binding Preferences database (http://cisbp.ccbr.utoronto.ca). To predict and identify LIN-39 binding motifs and UNC-3 binding motifs (identified in this paper) in the *cfi-1* enhancer (−11,391 bp to −12,146 bp), we utilized tools provided by MEME (Multiple Expectation maximization for Motif Elicitation) bioinformatics suite (http://meme-suite.org/), and performed Find Individual Motif Occurrences (FIMO) motif scanning analysis.

### Temporally controlled protein degradation

Temporally controlled protein degradation was achieved with the AID system (*Zhang et al., 2015*). TIR1 expression was driven by the ubiquitously active *eft-3* promoter in the transgene *ieSi57 [eft-3prom::tir1],* or a transgene that drives TIR1 selectively in neurons (*otTi28*). To induce degradation of proteins (LIN-39, CFI-1, UNC-3), the following alleles were used: *lin-39 (kas9 [lin-39::mNG::AID]), cfi-1 (kas16 [mNG::AID::cfi-1]),* and *unc-3 (ot837 [unc-3::mNG::AID]).* Worms at specific developmental stages (see figure legends for details) were grown at 20°C on NGM plates coated with 4 nM auxin (indole-3-acetic acid [IAA] dissolved in ethanol) or ethanol (negative control) for 1 day or 4 days before tested (see figure legends for more details). All plates were shielded from light.

### Worm tracking

Worms were maintained as mixed stage populations by chunking on NGM plates with *E. coli* OP50 as the food source. Worms were bleached and the eggs were allowed to hatch in M9 buffer to arrest as L1 larvae. L1 larvae were refed on OP50 and allowed to grow to day 2 of adulthood. On the day of tracking, five worms were picked from the incubated plates to each of the imaging plates (see below) and allowed to habituate for 30 min before recording for 15 min. Imaging plates are 35 mm plates with 3.5 mL of low-peptone (0.013% Difco Bacto) NGM agar (2% Bio/Agar, BioGene) to limit bacteria growth. Plates are stored at 4°C for at least two days before use. Imaging plates are seeded with 50 µl of a 1:10 dilution of OP50 in M9 the day before tracking and left to dry overnight with the lid on at room temperature.

### Behavioral feature extraction and analysis

All videos were analyzed using Tierpsy Tracker to extract each worm's position and posture over time (*Javer et al., 2018a*). These postural data were then converted into a set of behavioral features selected from a large set of features as previously described (*Javer et al., 2018b*). For each strain comparison, we performed unpaired two-sample t-tests independently for each feature. The false discovery rate was controlled at 5% across all strain and feature comparisons using the Benjamini Yekutieli procedure (*Benjamini and Yekutieli, 2001*).

### Statistical analysis

For data quantification, graphs show values expressed as mean ± standard error mean (SEM) of animals. The statistical analyses were performed using the unpaired t-test (two-tailed). Calculations were performed using the GraphPad QuickCalcs online software (http://www.graphpad.com/quickcalcs/). Differences with p<0.05 were considered significant.

## Acknowledgements

We thank the *Caenorhabditis* Genetics Center (CGC), which is funded by NIH Office of Research Infrastructure Programs (P40 OD010440), for providing strains. We thank the lab of Oliver Hobert for providing the OH14930 strain and Kaiyuan Tang for generating reporter gene constructs. We are grateful to Edwin Ferguson, Oliver Hobert, Daniele Canzio, Catarina Catela, and Weidong Feng for comments on this manuscript. This work was supported by a training grant [University of Chicago Initiative for Maximizing Student Development (IMSD), 2R25GM109439-06] to AO, a training grant (T32 GM007183) to EC, a Whitehall Foundation grant to PK, grants from National Institute of Neurological Disorders and Stroke (NINDS) of the NIH (Award Numbers R00NS084988 and R21NS108505) to PK, and a Medical Research Council grant MC-A658-5TY30 to AEXB.

# Additional information

## Funding

| Funder | Grant reference number | Author |
|---|---|---|
| National Institute of General Medical Sciences | Graduate student fellowship (2R25GM109439-06) | Anthony Osuma |
| National Institute of General Medical Sciences | Graduate student fellowship (T32 GM007183) | Edgar Correa |
| Whitehall Foundation | New Faculty Award (2017-12-50) | Paschalis Kratsios |
| National Institute of Neurological Disorders and Stroke | R00NS084988 | Paschalis Kratsios |
| National Institute of Neurological Disorders and Stroke | R21NS108505 | Paschalis Kratsios |
| Medical Research Council | MC-A658-5TY30 | André EX Brown |

The funders had no role in study design, data collection and interpretation, or the decision to submit the work for publication.

## Author contributions

Yinan Li, Conceptualization, Data curation, Formal analysis, Investigation, Visualization, Methodology, Writing - original draft, Writing - review and editing; Anthony Osuma, Edgar Correa, Formal analysis, Validation, Investigation; Munachiso A Okebalama, Pauline Dao, Olivia Gaylord, Jihad Aburas, Priota Islam, Investigation; André EX Brown, Project administration, Writing - review and editing; Paschalis Kratsios, Conceptualization, Supervision, Funding acquisition, Investigation, Writing - original draft, Project administration, Writing - review and editing

## Author ORCIDs

Paschalis Kratsios (iD) https://orcid.org/0000-0002-1363-9271

## Decision letter and Author response

Decision letter https://doi.org/10.7554/eLife.59464.sa1
Author response https://doi.org/10.7554/eLife.59464.sa2

# Additional files

## Supplementary files

• Supplementary file 1. Table summarizing UNC-3 ChIP-Seq signal distribution at the *cis*-regulatory regions of previously identified UNC-3 targets.

• Supplementary file 2. Table summarizing the results of protein class ontology analysis on novel target genes of UNC-3. In total, 1425 genes are classified into 25 protein classes.

• Supplementary file 3. UNC-3 binds to the *cis*-regulatory region of numerous genes expressed in cholinergic MNs.

• Transparent reporting form

## Data availability

Sequencing data have been deposited in GEO under accession code GSE143165. Moreover, all data generated or analysed during this study are included in the manuscript and supporting files.

The following dataset was generated:

| Author(s) | Year | Dataset title | Dataset URL | Database and Identifier |
|---|---|---|---|---|
| Li Y, Kratsios P | 2020 | ChIP-Sequencing data for the | https://www.ncbi.nlm. | NCBI Gene |

| transcription factor UNC-3/Ebf | nih.gov/geo/query/acc. cgi?acc=GSE143165 | Expression Omnibus, GSE143165 |

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
