## [Decision Letter]

**Acceptance summary:**

This paper addresses the important point of how transcription factors maintain the identity of the tissue they specify. In this case, the paper shows how UNC-3 maintains motor neuron identity and addresses whether it retains the same targets through time and how dynamic it is. The use of a ChIP time series allowed the identification of the gene targets in very few neurons, showing that UNC-3 binds to (not all) terminal genes and to other genes, engaging in a broad regulatory networks.

**Decision letter after peer review:**

Thank you for submitting your article "Establishment and maintenance of motor neuron identity via temporal modularity in terminal selector function" for consideration by *eLife*. Your article has been reviewed by three peer reviewers, and the evaluation has been overseen by a Reviewing Editor and Marianne Bronner as the Senior Editor. The following individual involved in review of your submission has agreed to reveal their identity: Chiou-Fen Chuang (Reviewer #2).

The reviewers have discussed the reviews with one another and the Reviewing Editor has drafted this decision to help you prepare a revised submission.

Your paper addresses the important point of how transcription factors maintain the identity of the tissue they specify. In this case, how does UNC-3 maintain motor neuron identity: Does it retain the same targets though time or is are they dynamic?

Your use of a ChIP time series allowed you to identify the gene targets in very few neurons, showing that UNC-3 binds to terminal genes and other genes, engaging in broad regulatory networks. You also showed, however, that some terminal genes might not always require direct UNC-3 input.

It is important that you clarify the difference between target genes for which UNC-3 works as a terminal selector versus those where it only initiates activation and is not required for maintenance.

The reviewers would therefore like you to examine whether UNC-3 is dispensable for reporter gene maintenance in some of the reporters. This should be a fairly easy experiment as you have UNC-3-AID worms where UNC-3 can be degraded after induction of gene expression. This point should also be discussed in detail in the paper.

Reviewer #1:

Understating how neurons keep their identity is an important question in developmental biology and neuroscience. Li and co-workers use a previously we-all established *C. elegans* model to investigate how UNC-3 maintains motor neuron identity. Specifically, the authors aim to address if UNC-3 maintains its targets though time or its binding and targets are dynamic.

The authors use an appropriate method, a ChIP time series. It is technically challenging to obtain a time series for a gene expressed in a few neurons, and thus by itself is an achievement. UNC-3 binds to "classic" terminal genes but also other regulatory genes. This data suggest UNC-3 is, and perhaps all terminal selectors, engage in broader regulatory networks. The UNC-3 binding dynamics also revealed that some terminal genes might not require direct UNC-3 input at some developmental stages. Moreover, the authors find a possible "initiation module" for the motor neuron program that required UNC-3 for maintenance.

I am less convinced the presented experiments support "Disruption of temporal modularity in UNC-3 function leads to locomotion defects", as the subtitle claims. The experiments are designed around the *cfi-1*. Moreover, "it is unclear whether the expression of UNC-3 targets from the "maintenance-only" module is critical for locomotion". Following the authors' logic, it is not clear UNC-3 is a terminal selector because it might not be required to maintain cell fate. I believe it is good to challenge hypotheses, but the experiments must be designed to do so.

The results are not itself conceptually novel as other temporal TF series have been described, even for motor neurons. However, the work is a significant addition to their previous *eLife* publication. Thus, this reviewer believes it is appropriate as a Research Advance.

Reviewer #2:

Establishment and maintenance of motor neuron identity via temporal modularity in terminal selector function.

In this paper, the authors identified direct targets of UNC-3/Ebf, the terminal selector of *C. elegans* cholinergic motor neurons, using ChIP-seq. They found that UNC-3 is required for the initiation and maintenance of expression of one group of target genes, but UNC-3 is only required to maintain expression of motor neuron-specific transcription factors such as CFI-1. The authors also found that the Hox proteins LIN-39 and MAB-5 act synergistically to control initiation and maintenance of *cfi-1* in cholinergic motor neurons through the same distal enhancer used by UNC-3 to maintain *cfi-1* expression. The authors used the auxin-inducible protein degradation system to show that *cfi-1* is required post-embryonically to maintain motor neuron subtype identity. In addition, a *cfi-1* null mutation and deletion of the *cfi-1* enhancer region cause locomotion defects. Furthermore, UNC-30, the terminal selector of GABAergic motor neurons, is required for the maintenance, but not initiation, of *cfi-1* expression in GABAergic neurons. In summary, this study reveals temporal modularity in terminal selector function for motor neuron identity.

The paper is well written and figures are clearly organized.

Reviewer #3:

The manuscript by Li et al. describes in depth analysis of UNC-3 target genes. The genes can be divided based on the UNC-3 role as initiation and maintenance genes and maintenance only. The authors focus on the regulation of one target transcription factor – *cfi-1*. They identify an enhancer that recapitulates to large extent normal *cfi-1* expression and provide evidence that the expression is initiated by Hox factors LIN-39 and MAB-5, while maintenance of the expression is controlled by UNC-3, possibly in conjunction with the Hox factors. Failure to maintain expression of CFI-1 in adult motor neurons has detrimental effects on motor neuron gene expression and worm motor behavior.

The findings are supported by extensive experimental evidence and in depth genetic and biochemical analysis of the enhancer. Overall this is an interesting in depth analysis of regulatory elements that sheds new light on dynamic changes in transcriptional complexes controlling target gene expression. Below I list several points that the authors should consider during the revision of the manuscript.

1) The authors place a lot of emphasis on division of target genes into two groups: initiation/ maintenance and maintenance only. However, the maintenance role in the first category has not been rigorously tested. That would require transient expression of *unc-3* early to turn the reporters on, followed by UNC-3 degradation. This raises the possibility that the first category can be subdivided into another subgroup – initiation only genes. Such possibility should be included in the Discussion.

2) The complexity of expression patterns observed with the transgenic reporters (some are on at L2 some only at L4, some are on in most neurons some in very few), demonstrates that the studied enhancers are regulated by multiple additional transcription factors besides UNC-3 (e.g. MAB-5/LIN-39 shown later in the manuscript). Thus, the conclusion that "UNC-3 can also act directly to either activate or repress expression of multiple TF-encoding genes" needs to be reconsidered, as a likely scenario is that UNC-3 binding to the regulatory element facilitates recruitment of another transcription factor that functions as a strong repressor.

3) Describing the results in Figure 4 the authors state: "Instead, it identified a 2.5 kb distal element (reporter #7) demarcated by the UNC-3 binding peak (Figure 4A)". Isn't this the same fragment as the one already tested in Figure 3? Please clarify.

4) The authors claim that 769bp sequence is necessary and sufficient for expression of reporter gene, however I do not see data testing the sufficiency of this particular fragment. Sufficiency is tested for construct #8 (552bp) that recapitulates only partially expression of *cfi-1* reporter. Moreover, it seems that even the full 2.5kb fragment is not sufficient to completely recapitulate the kinetics of *cfi-1* expression as the transgenic reporter is expressed in ~12 motor neurons (Figure 3B) in L2 while *cfi-1* endogenous reporter is on in ~25 neurons (Figure 4D).

5) Does UNC-3 bind to the *cfi-1* enhancer in the absence of MAB-5 and LIN-39 or do these factors have to "pioneer" the enhancer for UNC-3 recruitment? Would replacement of the enhancer with a synthetic element that carries only multimeric UNC-3 binding sites suffice to induce expression of *cfi-1* in cholinergic motor neurons?

6) What is the status of MAB-5 and LIN-39 expression in *unc-3* mutants? And how about UNC-3 expression in *lin-39*/*mab-5* mutants? The authors should consider the fate switch of motor neurons in *unc-3* mutants that could negatively affect maintenance of many motor neuron genes indirectly and independent of the UNC-3 binding sites identified in their promoters.

7) The 8 COE motifs mut animals should be examined for derepression of *glr-4* in adult motor neurons.

---

## [Author Response]

It is important that you clarify the difference between target genes for which UNC-3 works as a terminal selector versus those where it only initiates activation and is not required for maintenance.The reviewers would therefore like you to examine whether UNC-3 is dispensable for reporter gene maintenance in some of the reporters. This should be a fairly easy experiment as you have UNC-3-AID worms where UNC-3 can be degraded after induction of gene expression. This point should also be discussed in detail in the paper.

We thank the reviewers for their constructive comments. Through additional experiments and text modifications, we now clarify in the revised manuscript how UNC-3 controls over time these two distinct sets of target genes (module #1: terminal identity genes require UNC-3 for both initiation and maintenance; module #2: motor neuron-specific TFs require UNC-3 only for maintenance). All new data are included and discussed in detail in the revised manuscript (as described below).

Reviewer #1:[…]I am less convinced the presented experiments support "Disruption of temporal modularity in UNC-3 function leads to locomotion defects", as the subtitle claims. The experiments are designed around the cfi-1.

We totally agree with the reviewer and modified the subtitle in the Results to "*Minimal* disruption of temporal modularity in UNC-3 function leads to locomotion defects". As the reviewer correctly points out for this behavioral experiment, we only interfered with the ability of UNC-3 to maintain a single target gene (*cfi-1*) by using a CRISPR-engineered allele that bears mutations in the UNC-3 binding sites located at the distal enhancer of *cfi-1* (Figure 6). Apart from these specific mutations that selectively affect *cfi-1* maintenance (Figure 4E-F), animals carrying this allele do not bear any other mutations in UNC-3 targets either from module #1 (targets that require UNC-3 for initiation and maintenance) or module #2 (targets that require UNC-3 for maintenance only). Hence, the disruption of temporal modularity in UNC-3 function is minimal because UNC-3 can control all those targets, except one. In the revised manuscript, we modified the text in the Results (subsection “Minimal disruption of temporal modularity in UNC-3 function leads to locomotion defects”) to explain better our findings. Also, the phrase “*minimal* disruption…” appears in Abstract, Introduction, Results, and Discussion.

Moreover, "it is unclear whether the expression of UNC-3 targets from the "maintenance-only" module is critical for locomotion".

Following the reviewer’s suggestion, we modified this sentence to:

“However, it is unclear whether, *in cholinergic MNs*, the *maintained* expression of *any of the* UNC-3 targets from the “maintenance-only” module is critical for locomotion”.

Following the authors' logic, it is not clear UNC-3 is a terminal selector because it might not be required to maintain cell fate. I believe it is good to challenge hypotheses, but the experiments must be designed to do so.

We thank the reviewer for this comment, which enabled us to improve the manuscript in two ways. First, we used the auxin-inducible degradation (AID) system to selectively deplete UNC-3 in MNs at late larval/early adult stages. We found that UNC-3 is required to maintain the expression of 3 terminal identity genes (*unc-17/ VAChT, acr-2/AChR, glr-4/GluR*) (new Figure 3—figure supplement 2). In previous work (Kerk et al., 2017), we showed that UNC-3 is also required to maintain expression of another terminal identity gene (*unc-129/ TGFb*) in cholinergic MNs. In the current manuscript, we further found – by using a null allele – that UNC-3 is required to establish the early expression of all these 4 terminal identity genes (Figure 3A). Since the terminal selector definition entails that a TF is required to establish and maintain the expression of key effector molecules (e.g., neurotransmitter biosynthesis components, neurotransmitter receptors, ion channels) in a specific neuron type, our findings support the idea that UNC-3 operates as a terminal selector in cholinergic MNs. These new data are discussed in the Results (subsection “Temporal modularity of UNC-3 function in cholinergic motor neurons”) and Discussion (subsection “Temporal modularity offers key insights into how terminal selectors control neuronal identity over time”).

Second, we improved the Discussion (see the aforementioned subsection) by explaining that our findings do support the traditional view of how terminal selectors work. We find that module #1 targets do require UNC-3 both for initiation and maintenance, in accordance with the terminal selector concept. At the same time, our findings expand our understanding of terminal selectors because we suggest that terminal selector function can be more dynamic than previously thought. This is based on the data on module #2 targets that selectively require UNC-3 for maintenance (not initiation), suggesting terminal selectors can gain additional targets at later life stages.

The results are not itself conceptually novel as other temporal TF series have been described, even for motor neurons. However, the work is a significant addition to their previous eLife publication. Thus, this reviewer believes it is appropriate as a Research Advance.

To help the reader, we clarify in the revised manuscript the potential significance of temporal modularity in terminal selector function and how it expands our understanding of how terminal selectors control neuronal identity (Discussion, subsections “Temporal modularity in terminal selector function may represent a general principle for neuronal subtype diversity” and “Temporal modularity offers key insights into how terminal selectors control neuronal identity over time”).

Reviewer #3:[…] Below I list several points that the authors should consider during the revision of the manuscript.1) The authors place a lot of emphasis on division of target genes into two groups: initiation/ maintenance and maintenance only. However, the maintenance role in the first category has not been rigorously tested. That would require transient expression of unc-3 early to turn the reporters on, followed by UNC-3 degradation. This raises the possibility that the first category can be subdivided into another subgroup – initiation only genes. Such possibility should be included in the Discussion.

We thank the reviewer for this important point. We performed new experiments using the auxin-inducible degradation (AID) system to selectively deplete UNC-3 in MNs at late larval/early adult stages. We found that UNC-3 is required to maintain the expression of 3 terminal identity genes (*unc-17/ VAChT, acr-2/AChR, glr-4/GluR*) (new Figure 3—figure supplement 2). In previous work (Kerk et al., 2017), we showed that UNC-3 is also required to maintain expression of another terminal identity gene (*unc-129/ TGFb*) in cholinergic MNs. In the current manuscript, we further show – by using a null allele – that UNC-3 is required to establish the early expression of all these 4 genes (Figure 3A). Since the terminal selector definition entails that a TF is required to establish and maintain the expression of key effector molecules (e.g., neurotransmitter biosynthesis components, neurotransmitter receptors, ion channels) in a specific neuron type, our findings support the idea that UNC-3 operates as a terminal selector in cholinergic MNs. These new data are discussed in the Results (subsection “Temporal modularity of UNC-3 function in cholinergic motor neurons”) and Discussion (subsection “Temporal modularity offers key insights into how terminal selectors control neuronal identity over time”).

Moreover, the identification of UNC-3 targets from module #2 (e.g., UNC-3 is required to maintain, but not initiate, *cfi-1* expression) expand our understanding of how terminal selectors control neuronal identity over time (Discussion, subsection “Temporal modularity offers key insights into how terminal selectors control neuronal identity over time”). As the reviewer correctly points out, UNC-3 could also be controlling additional targets in a different manner, for example, by only inducing but not maintaining their expression. This possibility is now mentioned in the Discussion (subsection “Limitations of this study”).

2) The complexity of expression patterns observed with the transgenic reporters (some are on at L2 some only at L4, some are on in most neurons some in very few), demonstrates that the studied enhancers are regulated by multiple additional transcription factors besides UNC-3 (e.g. MAB-5/LIN-39 shown later in the manuscript). Thus, the conclusion that "UNC-3 can also act directly to either activate or repress expression of multiple TF-encoding genes" needs to be reconsidered, as a likely scenario is that UNC-3 binding to the regulatory element facilitates recruitment of another transcription factor that functions as a strong repressor.

We totally agree and have included this possibility in the Results (subsection “*Cis*-regulatory analysis reveals novel TFs as direct UNC-3 targets in motor neurons”). In addition, the section entitled “Hox proteins collaborate with stage-specific TFs to establish and maintain MN terminal identity” in the Discussion mentions that additional, yet-to-be-identified factors are involved to control expression of UNC-3 targets.

3) Describing the results in Figure 4 the authors state: "Instead, it identified a 2.5 kb distal element (reporter #7) demarcated by the UNC-3 binding peak (Figure 4A)". Isn't this the same fragment as the one already tested in Figure 3? Please clarify.

Yes, it is the same fragment. We changed this confusing sentence to the following: “Instead, it showed that the same 2.5 kb distal element (reporter #7), that drives RFP expression in subsets of *unc-3*-expressing cholinergic MNs (DA, DB, VA, VB subtypes) at larval and adult stages (Figure 3B), is also sufficient for embryonic (3-fold stage) MN expression (Figure 4A-C).”

4) The authors claim that 769bp sequence is necessary and sufficient for expression of reporter gene, however I do not see data testing the sufficiency of this particular fragment. Sufficiency is tested for construct #8 (552bp) that recapitulates only partially expression of cfi-1 reporter. Moreover, it seems that even the full 2.5kb fragment is not sufficient to completely recapitulate the kinetics of cfi-1 expression as the transgenic reporter is expressed in ~12 motor neurons (Figure 3B) in L2 while cfi-1 endogenous reporter is on in ~25 neurons (Figure 4D).

We apologize for the confusion. The revised manuscript now states: “We conclude that a distal 2.5kb enhancer (located ~12 kb upstream of *cfi-1*) is sufficient for *cfi-1* expression in nerve cord MNs. Genome editing suggests that a 769 bp sequence within this 2.5kb enhancer is required for both initiation and maintenance of *cfi-1* in nerve cord MNs (Figure 4D)”.

5) Does UNC-3 bind to the cfi-1 enhancer in the absence of MAB-5 and LIN-39 or do these factors have to "pioneer" the enhancer for UNC-3 recruitment? Would replacement of the enhancer with a synthetic element that carries only multimeric UNC-3 binding sites suffice to induce expression of cfi-1 in cholinergic motor neurons?

These are both excellent questions. The first one is technically very challenging as it would require us to perform an UNC-3 ChIP experiment on *lin-39 (-); mab-5 (-)* double mutant animals at different developmental stages (embryos and larvae). Getting the required amount of animals for this experiment is extremely difficult, judging from our lab’s experience with ChIP-seq. We also attempted to address this question genetically by increasing the expression levels of UNC-3 in cholinergic motor neurons of *lin-39 (-); mab-5 (-)* double mutants. However, we were unable to generate the required transgenic animals for this experiment, despite repeated attempts.

To address the second question, we generated transgenic animals carrying 5 copies of the UNC-3 binding site fused to GFP. These animals failed to show GFP expression in cholinergic motor neurons at every life stage examined (larvae, adults), suggesting that other factors are needed to cooperate with UNC-3 and activate its target genes. This scenario is in agreement with our *cfi-1* findings: we propose that the Hox proteins (LIN-39 and MAB-5) cooperate with UNC-3 to control *cfi-1* expression in cholinergic motor neurons (Figure 5G).

6) What is the status of MAB-5 and LIN-39 expression in unc-3 mutants? And how about UNC-3 expression in lin-39/mab-5 mutants?

In a previous study (Kratsios et al., 2017), we showed that the Hox proteins (LIN-39, MAB-5) and UNC-3 do not cross-regulate each other’s expression in cholinergic motor neurons. We added this important information in the Results (subsection “LIN-39 (Scr/Dfd/Hox4-5) and MAB-5 (Antp/Hox6-8) control *cfi-1* expression in cholinergic MNs through the same distal enhancer).

The authors should consider the fate switch of motor neurons in unc-3 mutants that could negatively affect maintenance of many motor neuron genes indirectly and independent of the UNC-3 binding sites identified in their promoters.

We are now mentioning this possibility in the Discussion (subsection “Limitations of this study”).

7) The 8 COE motifs mut animals should be examined for derepression of glr-4 in adult motor neurons.

As mentioned in the Results, CFI-1 represses *glr-4/GluR* expression in a subset of cholinergic MNs (~12 cells) that correspond to DA and DB subtypes. Animals carrying 8 COE motifs (UNC-3 sites) mutated in the context of the endogenous *cfi-1* reporter allele (*mNG::AID::cfi-1^8 COE motifs mut^*) show a reduction (from ~30 to ~15 *cfi-1+* cells), but not complete elimination, of *cfi-1* expression in adult (Day 1) cholinergic MNs (Figure 4E-F). This partial elimination is likely due to the presence of cryptic UNC-3 sites that we did not identify and therefore did not mutate in the *mNG::AID::cfi-1^8 COE motifs mut^* allele. Supporting this possibility, *cfi-1* expression at Day 1 in *unc-3 (-)* null animals is more severely affected compared to *mNG::AID::cfi-1^8 COE motifs mut^* animals (Figure 4E-F). Hence, the remaining expression of *cfi-1* in ~ 15 MNs of *mNG::AID::cfi-1^8 COE motifs mut^* animals is likely the reason that we did not observe *glr-4/GluR* ectopic expression in MNs of these animals (new data in Figure 5—figure supplement 2B). We mention this reason in the figure legend.

However, animals that carry the *mNG::AID::cfi-1^Δ enhancer (769 bp)^* allele and completely lack c*fi-1* expression in cholinergic motor neurons at all stages (Figure 4D) do show *glr-4/GluR* expression in MNs to similar levels observed in *cfi-1 (ot786)* putative null animals (new data in Figure 5—figure supplement 2B). We mention this data in the Results (subsection “*cfi-1*/Arid3a is required post-embryonically to maintain MN subtype identity”).